# Domain Adaptive Object Detection via Dynamic Causal Refinement

**Zeyu Ma** [1]   **Jiaqi Huang** [1]   **Yitong Qin** [1]   **Ziqiang Zheng** [* 1]
**Jiwei Wei** [1]   **Jie Zou** [1]   **Yang Yang** [1]   **Heng Tao Shen** [2]

## Abstract

Domain Adaptive Object Detection (DAOD) addresses the challenge of transferring object detectors from labeled source domains to unlabeled target domains. Existing domain adaptation methods primarily rely on feature distribution alignment, which enhances domain-invariant features (statistical invariance) but also inadvertently increases inherent domain-common spurious factors (e.g., common environmental contexts), which act as shortcut features rather than the true causal factors that determine generalization. We propose **Dynamic Causal Refinement (DCR)**, a novel framework that establishes a closed-loop feedback mechanism between data augmentation and model optimization to progressively refine causal features. Specifically, we design **Semantic Prediction Consistency (SPC)** to filter out domain-specific representation and learn robust statistical invariance, and **Discrepancy-Guided Causal Refinement (DGCR)** to actively suppress the dependence on domain-common spurious factors via spectral perturbation for causal refinement. This process encourages the detector to suppress its reliance on shortcut features and instead prioritize semantically meaningful causal representations. Extensive experiments on standard benchmarks demonstrate that our method outperforms state-of-the-art counterparts significantly. Our code is available on: https://github.com/Gakia457/DAOD-via-Dynamic-Causal-Refinement

## 1. Introduction

Modern object detectors achieve remarkable performance on in-domain data (Ren et al., 2015) but often struggle under distribution shifts, such as cross-weather (Cordts et al.,

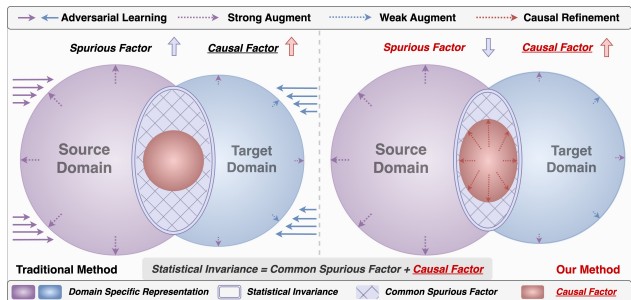

*Figure 1.* **Schematic Comparison.** The proposed method integrates causal refinement to suppress the dependence on domain-common spurious factors in statistically invariant features, thereby highlighting causal factors.

2016; Sakaridis et al., 2018) or cross-camera (Yu et al., 2020). Domain Adaptive Object Detection (DAOD) (Chen et al., 2018) addresses this challenge by transferring knowledge from a labeled source domain to an unlabeled target domain. The prevailing approaches in DAOD largely follow two paradigms: (1) feature alignment, which employs adversarial learning to minimize the discrepancy between domain representations (Ganin & Lempitsky, 2015; Saito et al., 2019), and (2) self-training, which iteratively refines pseudo-labels to align the target domain distribution with the source domain (Tarvainen & Valpola, 2017; Deng et al., 2021; Chen et al., 2022; Li et al., 2022).

Despite their distinct mechanisms, both approaches fundamentally pursue **statistical invariance**, which is often described as the learning domain-invariant feature, to indicate the alignment of cross-domain distributions. We argue that the statistical invariance pursued by current DAOD methods **does not equate to causal invariance**, as illustrated in Fig. 1 (left). Statistical invariance is achieved by aligning feature distributions, which encompass not only the causal factor that determines the object category but also unwanted domain-common spurious factors (i.e., common environmental contexts). In pursuit of more statistical invariance, models will exploit domain-common spurious factors (such as dominant backgrounds) as shortcuts (Geirhos et al., 2020), as these align more easily across domains than the causal factors from object features, such as detecting cars based on road texture. However, an over-reliance on domain-common spurious factors can undermine the model's capacity to learn the essential, causal features of the objects. We aim to

[1]University of Electronic Science and Technology of China [2]Tongji University. Correspondence to: Ziqiang Zheng <zhengziqiang1@gmail.com>.

*Proceedings of the $43^{rd}$ International Conference on Machine Learning*, Seoul, South Korea. PMLR 306, 2026. Copyright 2026 by the author(s).

identify predictors whose optimality stems from causal factors, not from spurious factors prevalent across domains. Therefore, we argue that the discrepancy between achieving statistical invariance and ensuring causal robustness is a primary reason for the performance ceiling in domain adaptation. We propose a novel dynamic causal refinement method to amplify the efficiency of causal factors and suppress domain-common spurious factor dependence within statistical invariance, as shown in Fig. 1 (right).

Given the challenge of disentangling the contextual dependency between causal content and spurious factors in pixel space, existing research in the frequency space (Huang et al., 2021; Xu et al., 2023) indicates that robust causal factors are primarily encoded in mid-frequency bands. In contrast, both high- and low-frequency components (Rahaman et al., 2019; Wang et al., 2020) are shown to contain mainly domain-specific spurious factors, while the transition bands between these frequency ranges contain domain-common spurious factors. However, a fundamental challenge remains: the division between high, intermediate, and low frequencies is inherently relative and qualitative, lacking a quantitative, principled criterion. Current methods for isolating causal bands use static filters or fixed bandwidths (Yang & Soatto, 2020; Xu et al., 2021), relying on a one-time, static data mapping followed by separate model training. We argue that a static spectral partitioning is suboptimal, as the boundary between causal and spurious factors is not fixed in high-dimensional feature space but evolves throughout the training process. Static partitioning restricts the model's maximum achievable performance. To achieve dynamic spectral partitioning, we propose a closed-loop **Dynamic Causal Refinement (DCR)** framework, specifically designed for DAOD.

Specifically, DCR is achieved through two synergistic components: **Semantic Prediction Consistency (SPC)** and **Discrepancy-Guided Causal Refinement (DGCR)**. First, SPC functions as an advanced self-training paradigm that aims to minimize distribution divergence more effectively for learning statistical invariance. The self-training process acts as a signal generator, utilizing the floating discrepancy between teacher and student outputs on target data as a performance-aware signal to guide the DGCR module. The DGCR module dynamically adjusts the bandpass range, linking spectral perturbation directly to the model's training state. Its adjustment rule balances the learning of statistical invariance and causal robustness. Specifically, a high teacher-student output discrepancy reduces spectral perturbation to prioritize learning statistically invariant features. Conversely, a low discrepancy permits stronger perturbation, further suppressing the dependence on domain-common spurious factors to highlight causal robustness. Consequently, spectral perturbation is transformed from a fixed preprocessing step into a responsive, integral component of the

adaptive training loop. Our approach ensures robust cross-domain predictions by iteratively suppressing the dependence on domain-common spurious factors and enabling dynamic causal refinement.

Our main contributions are summarized as follows:

- **Insight**: We rethink the DAOD problem by decomposing the notion of statistical invariance into two distinct components: domain-common spurious factors and causal factors. This refined conceptualization enables a more targeted learning strategy, leading to better performance.
- **Framework:** We propose a novel closed-loop framework where model training and data processing co-evolve. Breaking the paradigm where data mapping remains static during training, this approach utilizes the model's real-time feedback to dynamically regulate the data processing intensity and enables more robust representation learning.
- **Methodology:** We propose a Dynamic Causal refinement method, which is a feedback-driven mechanism where a semantic prediction consistency strategy ensures robust self-training and dynamically guides the causal refinement, actively suppressing the dependence on domain-common spurious factors and enhancing causal factors.

## 2. Related Work

### 2.1. Domain Adaptive Object Detection

Domain Adaptive Object Detection (DAOD) aims to generalize a detector trained on labeled source data to an unlabeled target domain. Existing methods can be broadly categorized based on their primary adaptation mechanism: adversarial learning and self-training. **Feature Alignment via Adversarial Learning.** Early approaches primarily focused on minimizing domain discrepancies through adversarial training. DA-Faster (Chen et al., 2018) proposed image-level and instance-level domain discriminators to align backbone and RoI features. SADA (Chen et al., 2021) extended this alignment to multi-scale FPN features, while MGA (Zhang et al., 2024) adopted multi-granularity alignment at pixel, instance, and category levels. More recently, DA2OD (He et al., 2025b) proposed a differential alignment strategy that dynamically reweights instances based on prediction discrepancy and balances foreground-background adaptation intensities. **Self-Training and Consistency Learning.** Currently, the dominant paradigm is self-training based on the Mean Teacher framework (Tarvainen & Valpola, 2017; Deng et al., 2021), where an EMA-updated teacher generates pseudo-labels to supervise the student. To enhance the robustness of this framework, many methods integrate auxiliary consistency or alignment objectives. AT (Li et al., 2022) utilized a weak-strong augmentation strategy to ensure reliable pseudo-labels generation, complemented by auxiliary adversarial feature alignment. MIC (Hoyer et al., 2023) combined SADA alignment with

a Masked Image Consistency module to enforce prediction consistency under heavy occlusion. SSAL (Munir et al., 2023) coordinated adversarial alignment and self-training by leveraging prediction uncertainty to filter samples for different objectives. Other self-training approaches focus on refining the pseudo-labels mechanism itself. PT (Chen et al., 2022) proposed a threshold-free paradigm that leverages probabilistic modeling to capture prediction uncertainty. CAT (Kennerley et al., 2024) exploited inter-class dynamics to guide tailored augmentations for mitigating class bias. DT (Lavoie et al., 2025) leveraged vision foundation models to generate high-quality pseudo-labels for robust student supervision. The self-training paradigm is built on pseudo-labels, which collapse the informative prediction distribution into deterministic assignments, discarding other valuable distribution information. Moreover, consistency constraints are often applied only to model outputs, overlooking the rich semantic information present in intermediate feature representations. In contrast, our method leverages both the whole prediction distribution and intermediate semantic features to enhance the performance of Domain Adaptive Object Detection.

## 2.2. Causal Learning

Domain Adaptive Object Detection (DAOD) aims to learn statistically invariant representations between a labeled source domain and an unlabeled target domain. Building on the causal and spurious correlation from the IR (Chang et al., 2020), we further refine spurious correlation by subdividing it into domain-specific and domain-common spurious factors. Statistical invariance is formed by the combination of causal factors and domain-common spurious factors, which undermines robust adaptation. Existing object detection research on causal feature learning can be broadly categorized into two paradigms: invariance learning and active perturbation. The former, typified by IRM-style formulations (Arjovsky et al., 2019), enforced a shared predictor across multiple domains to recover invariant causal mechanisms. However, these methods require sufficiently diverse domains and, in the single-source domain adaptive object detection setting, learned features are merely statistically invariant rather than truly causal. The latter paradigm explicitly intervenes on data or features to test the stability of performance under perturbation (Xu et al., 2021; Huang et al., 2021). FDA (Yang & Soatto, 2020) swapped low-frequency components between source and target images to remove nuisance factors that are difficult to isolate in pixel space, while MAD (Xu et al., 2023) injected Gaussian perturbations into high- and low-frequency bands and contrasts multiple spectral views to isolate features that are both invariant and causal. These approaches are their reliance on fixed perturbation schedules, which fail to adapt to the model's evolving internal state. In contrast, we design a discrepancy-guided dynamic scheduling mechanism for frequency-space perturbations. By adapting the perturbation strategy based on the teacher-student discrepancy, our method maintains focus on causal factors during training.

## 3. Method

### 3.1. Further Insight for DAOD

Existing DAOD approaches primarily follow two paradigms: adversarial learning and self-training. The former explicitly aligns feature distributions by making extracted features indistinguishable to a domain discriminator (Further analysis in Sec. C.1) , while the latter implicitly aligns feature distributions by utilizing high-confidence pseudo-labels to pull target domain features toward source domain features. Regardless of their methodological differences, they ultimately pursue domain invariance. We identify that the optimization goal is essentially domain statistical invariance, which represents the feature intersection between source and target domains. In Fig. 1, such statistical invariance contains not only causal factors that truly determine generalization but also spurious factors, which are common across domains yet serve only to reduce training loss. To suppress these spurious factors and highlight causal factors, we propose the Dynamic Causal Refinement (DCR) framework in Fig. 2. Specifically, DCR employs **Semantic Prediction Consistency (SPC)** to align the distribution between source and target domains, and establish a robust statistical invariance, and **Discrepancy-Guided Causal Refinement (DGCR)** to actively suppress the dependence on spurious factors via spectral perturbation for further causal refinement.

### 3.2. Semantic Prediction Consistency

Following the Mean Teacher paradigm (Tarvainen & Valpola, 2017), we employ a teacher-student self-training framework. On the labeled source domain $\mathcal{D}_S$, the student network $\theta_{stu}$ is optimized solely via the standard supervised detection loss $\mathcal{L}_{sup}$, **free from any auxiliary component**. Meanwhile, we propose Semantic Prediction Consistency (SPC) to drive adaptation **exclusively on the unlabeled target domain $\mathcal{D}_T$**. Here, the teacher network $\theta_{tch}$ is updated as an Exponential Moving Average (EMA) of the student.

Current self-training methods (Li et al., 2022; Kennerley et al., 2024; Lavoie et al., 2025) predominantly select pseudo-labels rashly based on maximum confidence, which introduces false positives and ignores the statistical properties of the prediction distribution. Moreover, this exclusive reliance on pseudo-labels fails to leverage rich intermediate representations. We design SPC to enforce consistency between the teacher and student frameworks for adequately learning the target domain data via two complementary strategies: **Semantic Context Consistency** and **Prediction Distribution Consistency** (More Details in Sec.A.1).

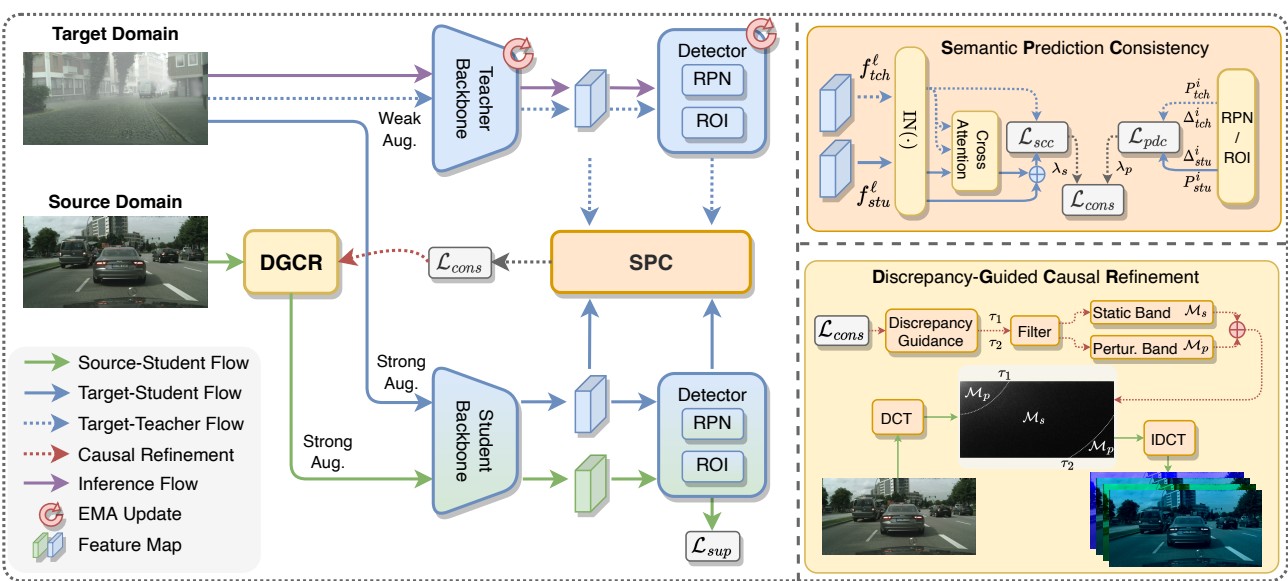

*Figure 2.* **Overview of The Dynamic Causal Refinement (DCR) Framework**. **Training Stage:** Source images undergo DGCR and strong augmentation for supervised student learning. Target images (with strong and weak augmentations) are fed into both student and teacher models to generate features and predictions for the SPC module to compute semantic prediction loss $\mathcal{L}_{cons}$, which provides feedback to dynamically guide the DGCR module. **Inference Stage:** Only the teacher model is deployed without any extra components.

**Semantic Context Consistency.** Previous self-training methods rely solely on pseudo-labels from the teacher-student framework, overlooking the potential of mid-level features for consistency learning. So, we identify channel-wise statistics (e.g., mean and variance) as the style information (Huang & Belongie, 2017), which are parts of domain-specific spurious factors. Therefore, given a feature map $f^\ell \in \mathbb{R}^{C \times H \times W}$ from layers $\ell \in \mathcal{P}$ in FPN, we apply Instance Normalization (IN) to remove style statistics:

$$\hat{f}^\ell = \text{IN}(f^\ell) = \frac{f^\ell - \mu(f^\ell)}{\sigma(f^\ell)}, \tag{1}$$

where $\mu, \sigma$ represent spatial statistics over $(H, W)$.

With style statistics removed, the feature representations can further achieve statistical invariance. Then, we flatten the normalized maps $\hat{f}^\ell_{stu}$ and $\hat{f}^\ell_{tch}$ into sequences $X^\ell_{stu}$ and $X^\ell_{tch}$. The student acts as queries $Q^\ell_{stu}$ to aggregate contextual guidance from the teacher, which serves as keys $K^\ell_{tch}$ and values $V^\ell_{tch}$. The attention map is derived using a layer-dependent temperature $\tau_\ell$:

$$A^\ell = \text{Softmax}\left(\frac{Q^\ell_{stu}(K^\ell_{tch})^\top}{\tau_\ell \sqrt{d_k}}\right) V^\ell_{tch}, \tag{2}$$

where $d_k$ denotes the projection dimension. To prevent feature degradation, we incorporate a residual connection from the input student sequence with a scaling factor $\lambda$:

$$H^\ell = A^\ell + \lambda X^\ell_{stu}. \tag{3}$$

Finally, we apply global average pooling $g(\cdot)$ to condense both the guidance-enriched student sequence $H^\ell$ and the

teacher's feature sequence $X^\ell_{tch}$. We maximize the cosine similarity $s^\ell$ between them, explicitly enforcing the consistency in the global presence of semantic features:

$$s^\ell = \frac{g(H^\ell)^\top g(X^\ell_{tch})}{\|g(H^\ell)\|_2 \|g(X^\ell_{tch})\|_2}, \tag{4}$$

$$\mathcal{L}_{scc} = \sum_{\ell \in \mathcal{P}} \alpha^\ell \max\left(0, \ 1 - m^\ell - s^\ell\right), \tag{5}$$

where $\alpha^\ell$ is the balancing weight for level $\ell$, and $m^\ell$ is the margin. By back-propagating gradients through the learnable attention projections, it explicitly compels the student backbone to learn style-invariant and semantically enriched representations. This mechanism enhances robustness against domain shifts with **zero inference overhead**. Note that we didn't utilize these constrained features for pseudo-labels generation.

**Prediction Distribution Consistency.** Having established semantic context consistency learning between teacher and student networks in the target domain, we proceed to address the shortcomings of existing methods. Firstly, they collapse the whole category probability distribution into a deterministic category, but discarding valuable prediction distribution and exacerbating the risk of **false positives**. Secondly, the issue of location errors in pseudo-labels has been largely overlooked. (Further analysis in Sec B.5) To mitigate these problems, we introduce a novel PDC component. Its design is based on two key operations: universal anchor sampling and the sharing of proposals generated by the student network, thereby ensuring that category prediction distribution consistency is computed with fewer location errors.

Formally, let $\mathcal{S} = \mathcal{S}_{rpn} \cup \mathcal{S}_{roi}$ be the union set containing all shared anchors and proposals. For each $i \in \mathcal{S}$, let $P_{(\cdot)}^i$ denote the predicted category distribution, and $\Delta_{(\cdot)}^i$ denote the predicted box offsets used to refine the anchors/proposals. We minimize the KL divergence $\mathcal{D}_{\mathrm{KL}}$ for classification and the Smooth L1 distance $\| \cdot \|$ for regression:

$$\mathcal{L}_{pdc} = \frac{1}{|\mathcal{S}|} \sum_{i \in \mathcal{S}} \left( \mathcal{D}_{\mathrm{KL}}\big(P_{tch}^i \| P_{stu}^i\big) + \mathbb{I}_{fg} \cdot \big\| \Delta_{stu}^i - \Delta_{tch}^i \big\| \right), \tag{6}$$

where $\mathbb{I}_{fg}$ indicates whether the teacher classifies the $i$-th candidate as foreground.

**Consistency Loss.** We integrate the semantic and prediction consistency terms into a unified loss:

$$\mathcal{L}_{cons} = \lambda_s \mathcal{L}_{scc} + \lambda_p \mathcal{L}_{pdc}, \tag{7}$$

where $\lambda_s$ and $\lambda_p$ are balancing hyperparameters.

Crucially, we treat $\mathcal{L}_{cons}$ not merely as an optimization objective, but as a real-time measure of the distribution discrepancy between the teacher and student networks on target domain data. We argue that this discrepancy dynamically reflects the learning of domain-invariant causal factors. Therefore, we feed this raw discrepancy signal into the causal refinement module (Sec. 3.3) to dynamically modulate the source domain frequency perturbation.

### 3.3. Discrepancy-Guided Causal Refinement

As discussed in Sec. 1, statistical invariance does not equate to causal invariance. Although domain-specific spurious factors dominate the frequency extremes and causal factors occupy the mid-band, the core challenge lies in the junction zones. Here, domain-common spurious factors are inextricably entangled with causal factors. Since explicitly decoupling causal factors is intractable, we design a dynamic inverse adjust strategy and continuously perturb the spurious-factor-dense regions (corresponding to high- and low-frequency bands) while protecting a middle band to refine causal features. The cut-off boundaries for these perturbations are not fixed; they fluctuate adaptively within the junction zones between bands, guided by the model's learning signal. This mechanism compels the model to learn invariant causal features amid varying spurious correlations. To implement this strategy, we design the **Discrepancy-Guided Causal Refinement (DGCR)** component to refine the causal factors within the learned statistical invariance.

**Spectral Perturbation Mechanism.** Specifically, let $x \in \mathbb{R}^{H \times W \times C}$ be the input image, transformed into the frequency space via the Discrete Cosine Transformation (DCT) as $\mathcal{F}(x)$. We employed two constrained adaptive cutoff thresholds $\tau_1, \tau_2$ ($\tau_1 < \tau_2$) to partitioning the spectrum into two distinct regions: a **static band** (mid-frequency) and

a **perturbation band** (high- and low-frequency). These regions are formally defined by the masks $\mathcal{M}_s$ and $\mathcal{M}_p$:

$$\mathcal{M}_s(u, v) = \mathbb{I}_{[\tau_1 \leq \rho(u,v) \leq \tau_2]}, \tag{8}$$

$$\mathcal{M}_p(u, v) = \mathbf{1} - \mathcal{M}_s(u, v), \tag{9}$$

where $\rho(u, v)$ represents the radial frequency coordinate, $\mathbf{1}$ denotes the all-ones matrix, and $\mathbb{I}_{[\cdot]}$ is the indicator function. To implement the inverse strategy, we inject Gaussian noise $\varepsilon \sim \mathcal{N}(0, \sigma^2)$ exclusively into the perturbation band:

$$\tilde{x} = \mathcal{F}^{-1}\Big(\mathcal{M}_s \odot \mathcal{F}(x) + \mathcal{M}_p \odot (\mathcal{F}(x) + \varepsilon)\Big), \tag{10}$$

where $\odot$ denotes the element-wise product.

**Discrepancy Dynamic Guidance.** The bandwidth $[\tau_1, \tau_2]$ critically determines the intensity of the perturbation. We regulate it dynamically using the consistency loss $\mathcal{L}_{cons}$ (derived in Sec. 3.2) as a feedback signal. $\mathcal{L}_{cons}$ quantifies the distribution discrepancy between the teacher and student network on target domain data. A high discrepancy implies the model has not yet mastered basic statistical invariance. Conversely, a low discrepancy suggests that the model is approaching statistical invariance. Without external guidance, the model is misled into adopting domain-common factors as convenient shortcuts, achieving statistical invariance quickly but failing to learn the true causal structure.

Given the stochastic fluctuations inherent in batch-wise training, we employ an Exponential Moving Average (EMA) with momentum $\beta \in [0, 1]$ to update the historical baseline $\bar{\mathcal{L}}^{(t)}$ and standardize the instantaneous loss $\mathcal{L}_{cons}^{(t)}$ at iteration $t$ into a relative deviation $\delta^{(t)}$:

$$\bar{\mathcal{L}}^{(t)} = \beta \bar{\mathcal{L}}^{(t-1)} + (1 - \beta)\mathcal{L}_{cons}^{(t)}, \tag{11}$$

$$\delta^{(t)} = \frac{\mathcal{L}_{cons}^{(t)} - \bar{\mathcal{L}}^{(t)}}{\bar{\mathcal{L}}^{(t)} + \xi}. \tag{12}$$

Here, $\xi$ is a small constant for numerical stability, and $\delta^{(t)}$ serves as a normalized indicator of whether the current adaptation is in a high-discrepancy or high-consistency trend relative to its history. To convert this unbounded deviation into a stable control signal, we map it to a modulation factor $\alpha_t$ via a hyperbolic tangent function:

$$\alpha_t = 1 + \eta \tanh(\kappa \cdot \delta^{(t)}). \tag{13}$$

Here, $\kappa$ is a sensitivity hyperparameter that scales the response to deviation changes, and $\eta$ is an amplitude factor that sets the maximum modulation range. By design, the $\tanh(\cdot)$ function restricts $\alpha_t$ within the bounded interval $[1 - \eta, 1 + \eta]$. This mechanism prevents extreme frequency shifts that could destabilize training, ensuring smooth and adaptive guidance based on the current discrepancy trend.

*Table 1.* **Comparison (%) Results on Adaptation from Cityscapes to Foggy Cityscapes (C → F) and Sim10k to Cityscapes (S → C).** The vertical double line separates the two distinct tasks. **Bold** and underline denote the best and second-best adaptation results, respectively. The results for ALDI++ on C → F are based on reproduction using the official code. AT* results on Foggy (0.02) are taken from MRT(Zhao et al., 2023). "-" denotes inapplicable entries.

| Method | Split | C → F | | | | | | | | | S → C |
| | | Person | Rider | Car | Truck | Bus | Train | Motor | Bicycle | mAP | AP50$_{car}$ |
|---|---|---|---|---|---|---|---|---|---|---|---|
| PT(Chen et al., 2022) *[ICML'22]* | | 43.2 | 52.4 | 63.4 | 33.4 | 56.6 | 37.8 | 41.3 | 48.7 | 47.1 | - |
| AT(Li et al., 2022) *[CVPR'22]* | | 45.5 | 55.1 | 64.2 | 35.0 | 56.3 | 54.3 | 38.5 | 51.9 | 50.9 | - |
| DA-pro(Li et al., 2023a) *[NIPS'23]* | | 55.4 | 62.9 | 70.9 | 40.3 | 63.4 | 54.0 | 42.3 | 58.0 | 55.9 | 62.9 |
| REACT(Li et al., 2024a) *[TIP'24]* | | 51.4 | 57.9 | 67.4 | 37.7 | 58.4 | 52.8 | 44.6 | 54.6 | 53.1 | 58.6 |
| DA-ADA(Li et al., 2024b) *[NIPS'24]* | ALL | 57.8 | 65.1 | 71.3 | 43.1 | 64.0 | 58.6 | 48.8 | 58.7 | 58.5 | 67.3 |
| FGPro(Wen et al., 2025) *[TCYB'25]* | | 46.2 | 54.8 | 65.6 | 38.9 | 63.1 | 47.9 | 44.7 | 50.4 | 51.4 | 65.7 |
| DT(Lavoie et al., 2025) *[CVPR'25]* | | 48.5 | 60.0 | 65.4 | 47.2 | 66.5 | 52.9 | 46.2 | 56.7 | 55.4 | - |
| SEEN-DA(Li et al., 2025) *[CVPR'25]* | | 58.5 | 64.5 | 71.7 | 42.0 | 61.2 | 54.8 | 47.1 | 59.9 | 57.5 | 66.8 |
| ALDI++(Kay et al., 2025) *[TMLR'25]* | | 65.6 | 69.4 | 77.4 | 48.4 | 72.3 | 57.8 | 54.7 | 66.1 | 64.0 | - |
| **Ours** | ALL | **70.2** | **73.7** | **79.6** | **53.4** | **75.8** | **60.5** | **63.3** | **68.8** | **68.2** | - |
| DA-Faster(Chen et al., 2018) *[CVPR'18]* | | 25.0 | 31.0 | 40.5 | 22.1 | 35.3 | 20.2 | 20.0 | 27.1 | 27.6 | 39.0 |
| PT(Chen et al., 2022) *[ICML'22]* | | 40.2 | 48.8 | 59.7 | 30.7 | 51.8 | 30.6 | 35.4 | 44.5 | 42.7 | 55.1 |
| AT*(Chen et al., 2022) *[CVPR'22]* | | 43.7 | 54.1 | 62.3 | 31.9 | 54.4 | 49.3 | 35.2 | 47.9 | 47.4 | - |
| SIGMA++(Li et al., 2023b) *[TPAMI'23]* | | 46.4 | 45.1 | 61.0 | 32.1 | 52.2 | 44.6 | 34.8 | 39.9 | 44.5 | 57.7 |
| MIC(Hoyer et al., 2023) *[CVPR'23]* | | 50.9 | 55.3 | 67.0 | 33.9 | 52.4 | 33.7 | 40.6 | 47.5 | 47.6 | 58.9 |
| NSA-UDA(Zhou et al., 2023) *[ICCV'23]* | | - | - | - | - | - | - | - | - | 52.7 | 56.3 |
| MGA(Zhang et al., 2024) *[TPAMI'24]* | 0.02 | 47.9 | 50.1 | 64.9 | 34.8 | 58.0 | 45.6 | 38.3 | 43.7 | 47.9 | 55.8 |
| MTM(Weng & Yuan, 2024) *[AAAI'24]* | | 51.0 | 53.4 | 67.2 | 37.2 | 54.4 | 41.6 | 38.4 | 47.7 | 48.9 | 58.1 |
| SOCCER(Cui et al., 2024) *[MM'24]* | | 51.7 | 57.7 | 68.6 | 38.2 | 51.6 | 47.5 | 41.6 | 51.7 | 51.1 | 63.8 |
| DSD-DA(Feng et al., 2024) *[ICML'24]* | | 49.1 | 59.3 | 66.2 | 35.8 | 60.0 | 47.1 | 45.2 | 54.9 | 52.2 | 52.5 |
| CAT(Kennerley et al., 2024) *[CVPR'24]* | | 44.6 | 57.1 | 63.7 | 40.8 | 66.0 | 49.7 | 44.9 | 53.0 | 52.5 | - |
| FGT(He et al., 2025a) *[ICCV'25]* | | 55.6 | 63.1 | 71.7 | 41.1 | 56.9 | 48.9 | 49.1 | **66.2** | 56.6 | - |
| DA2OD(He et al., 2025b) *[AAAI'25]* | | 59.8 | 62.8 | 73.7 | 40.3 | 59.4 | 56.1 | 47.8 | 58.3 | 57.3 | 69.7 |
| ALDI++(Kay et al., 2025) *[TMLR'25]* | | 64.2 | 64.7 | 76.4 | 42.1 | 63.8 | 55.3 | 52.5 | 60.3 | 59.9 | 69.1 |
| **Ours** | 0.02 | **69.6** | **71.8** | **79.0** | **51.3** | **71.5** | **59.6** | **59.2** | 63.0 | **65.6** | **73.9** |
| Oracle | | 74.6 | 70.9 | 82.8 | 48.3 | 62.4 | 60.7 | 55.7 | 64.1 | 64.9 | 84.9 |

To operationalize this strategy, we employed an Elastic Spectral Boundary mechanism. Using initialized anchor thresholds $[\tau_1^0, \tau_2^0]$, we update the current cutoff frequencies $[\tau_1^{(t)}, \tau_2^{(t)}]$ by applying the modulation factor $\alpha_t$ directly:

$$\tau_1^{(t)} = \tau_1^0 \cdot \alpha_t^{-1}, \quad \tau_2^{(t)} = \tau_2^0 \cdot \alpha_t. \quad (14)$$

This inverse scaling formulation ensures that a single control variable $\alpha_t$ synchronously regulates the bandwidth.

This mechanism implements a discrepancy-dynamic spectral perturbation with two distinct phases:

- **Stable Phase ($\alpha_t > 1$):** When the discrepancy signal is high, the static band widens to ensure learnability. It prioritizes the retention of causal factors to prevent performance degradation. We tolerate the inclusion of some domain-common spurious factors as necessary compromises to ensure that statistical invariance is prioritized.
- **Refining Phase ($\alpha_t < 1$):** When statistical invariance is established, the static band narrows while the perturbation band expands. This forces the model to learn core causal factors, ensuring robust performance by aggressively suppressing potential domain-common spurious factors.

This dynamic scheduling adaptively regulates spectral perturbations, guiding the model to learn from solely domain-invariant features to causal factor refinement.

## 4. Experiments

### 4.1. Datasets and Implementation Details

**Dataset**. We evaluate our method on three cross-domain scenarios: **C→F** for cross-weather adaptation, **C→B** for cross-camera adaptation, and **S→C** for synthetic-to-real adaptation. For the **C→F** experiment, we report results on both the **ALL** split (0.005, 0.01, and 0.02) and the hardest **0.02** split. More details are provided in Sec. A.3.

**Implementation Details**. We use Faster R-CNN with ResNet-50 (He et al., 2016) and FPN (Lin et al., 2017), initialized via RegionCLIP (Zhong et al., 2022) (More Details in Sec. B.2). The inference model does not introduce any additional parameters (More Details in Sec. C.2). All implementations are based on Detectron2. We report the mean Average Precision at an IoU threshold of 0.5 (mAP@50) across all benchmarks. Further details regarding parameters, data augmentations, and experimental settings are available in Sec. A.2 and Sec. A.3.

*Table 2.* **Comparison (%) Results on Adaptation from Cityscapes to BDD100k Daytime (C → B). Bold** and underline denote the best and second-best adaptation results, respectively. "-" denotes that the original paper does not report the results.

| Method | Person | Rider | Car | Truck | Bus | Motor | Bicycle | mAP |
|---|---|---|---|---|---|---|---|---|
| DA-Faster(Chen et al., 2018) *[CVPR'18]* | 28.9 | 27.4 | 44.2 | 19.1 | 18.0 | 14.2 | 22.4 | 24.9 |
| PT (Chen et al., 2022) *[ICML'22]* | 40.5 | 39.9 | 52.7 | 25.8 | 33.8 | 23.0 | 28.8 | 34.9 |
| SIGMA++(Li et al., 2023b) *[TPAMI'23]* | 47.5 | 30.4 | 65.6 | 21.1 | 26.3 | 17.8 | 27.1 | 33.7 |
| NSA-UDA(Zhou et al., 2023) *[ICCV'23]* | - | - | - | - | - | - | - | 35.5 |
| MIC(Hoyer et al., 2023) *[CVPR'23]* | 49.8 | 36.8 | 68.1 | 24.0 | 25.6 | 18.7 | 30.7 | 36.2 |
| REACT(Li et al., 2024a) *[TIP'24]* | - | - | - | - | - | - | - | 35.8 |
| CAT(Kennerley et al., 2024) *[CVPR'24]* | 44.6 | 41.5 | 61.2 | 31.4 | 34.6 | 24.4 | 31.7 | 38.5 |
| SOCCER(Cui et al., 2024) *[MM'24]* | 56.8 | 42.2 | 73.1 | 31.0 | 29.5 | 26.1 | 33.8 | 41.8 |
| DA2OD(He et al., 2025b) *[AAAI'25]* | 61.4 | 45.4 | 75.4 | 33.0 | 36.2 | 29.5 | 36.7 | 45.8 |
| DT(Lavoie et al., 2025) *[CVPR'25]* | 51.6 | 47.0 | 66.6 | **44.3** | **45.9** | 38.3 | 40.8 | 47.8 |
| **Ours** | **65.0** | **55.1** | **78.4** | 43.7 | 40.0 | **44.2** | **43.0** | **52.8** |
| Oracle | 72.9 | 58.1 | 86.6 | 65.9 | 66.1 | 49.1 | 54.9 | 64.8 |

## 4.2. Comparison With State-of-the-Arts

**Cityscapes → Foggy Cityscapes.** Tab. 1 (left) reports results on cross weather adaptation. Our method achieves **65.6%** mAP on the 0.02 split, outperforming all existing methods in Tab. 1, including the recent state-of-the-art ALDI++, in both overall mAP and almost per-class AP. On the "ALL" split, our method further boosts the performance to **68.2%** mAP, maintaining a significant lead over ALDI++ with 64.0% mAP. Our method achieves the highest AP in all categories on Cityscapes. Compared to DT (Lavoie et al., 2025) that used large pretrained models DINOv2 (Oquab et al., 2024) to generate pseudo-labels, our approach avoids a heavy computational burden during training. Even with a smaller model architecture, our method achieves superior accuracy. In Sec. C.3, we further compare the proposed method with the recent segmentation foundation model SAM 3 (Carion et al., 2025) and other approaches that directly apply VLM to perform cross-domain inference. While SAM 3 demonstrates impressive performance, our method maintains superior results. We also present a comparison of inference speed with these large-scale models in Tab. 15, where our approach shows a significant advantage. These results validate our causal refinement as a novel and effective paradigm for domain adaptive object detection, establishing a foundation for sustained performance improvements.

**Sim10K → Cityscapes.** Tab. 1 (right) reports the results on synthetic-to-real adaptation. Our method obtains 73.9% AP50, significantly improving over the previous best DA2OD result of 69.7% by **4.2%** and establishing a new state of the art on this benchmark.

**Cityscapes → BDD100K Daytime.** Tab. 2 presents the results for this cross-camera challenging scenario. Even under such difficult conditions, our method achieves **52.8%** mAP, significantly surpassing the previous best method DT, at 47.8% mAP by a solid margin of 5.0%. In detail, our method ranks first in five of the seven categories and attains a strong second place in the remaining "Truck" and "Bus" categories, underscoring its overall leading performance. Given a larger target dataset, we demonstrate that causal refinement in base framework substantially outperforms VFM based method such as DT (Lavoie et al., 2025).

*Table 3.* **Overall Component Design.** Ablation study on the effectiveness of components: R-Init, SPC, and DGCR.

| R-Init | SPC | DGCR | mAP | Gains |
|---|---|---|---|---|
| - | - | - | 59.1 | - |
| ✓ | - | - | 62.3 | +3.2 |
| ✓ | ✓ | - | 64.6 | +5.5 |
| ✓ | - | ✓ | 63.9 | +4.8 |
| ✓ | ✓ | ✓ | **65.6** | **+6.5** |

*Table 4.* **Internal Component Design.** Ablation study on (a) SCC components and (b) DGCR spectral perturbation strategies. "R." denotes Random and "F." denotes Fixed.

| (a) SCC Internal Design | | | | | |
|---|---|---|---|---|---|
| **Variant** | None | w/o IN | w/o Attn. | w/o Resi. | **Ours** |
| mAP | 63.0 | 63.3 | 63.2 | 64.0 | **64.6** |

| (b) DGCR Internal Design | | | | | |
|---|---|---|---|---|---|
| **Variant** | None | F. Band | R. Band | R. Factor | **Ours** |
| mAP | 64.6 | 64.9 | 64.8 | 64.5 | **65.6** |

## 4.3. Ablation Study

**Ablation of Overall Component Design.** Tab. 3 evaluates the contributions of RegionCLIP initialization (R-Init), Semantic Prediction Consistency (SPC), and Discrepancy-Guided Causal Refinement (DGCR). The standard ImageNet-initialized model yields 59.1% mAP. Enabling R-Init alone boosts the performance to 62.3%, confirming the benefit of strong initialization. Building on R-Init, adding SPC or DGCR yields 64.6% and 63.9% mAP, respectively. It is important to note that in the ablation where the SPC component is removed, raw guidance is provided exclusively by the original pseudo-labels loss.

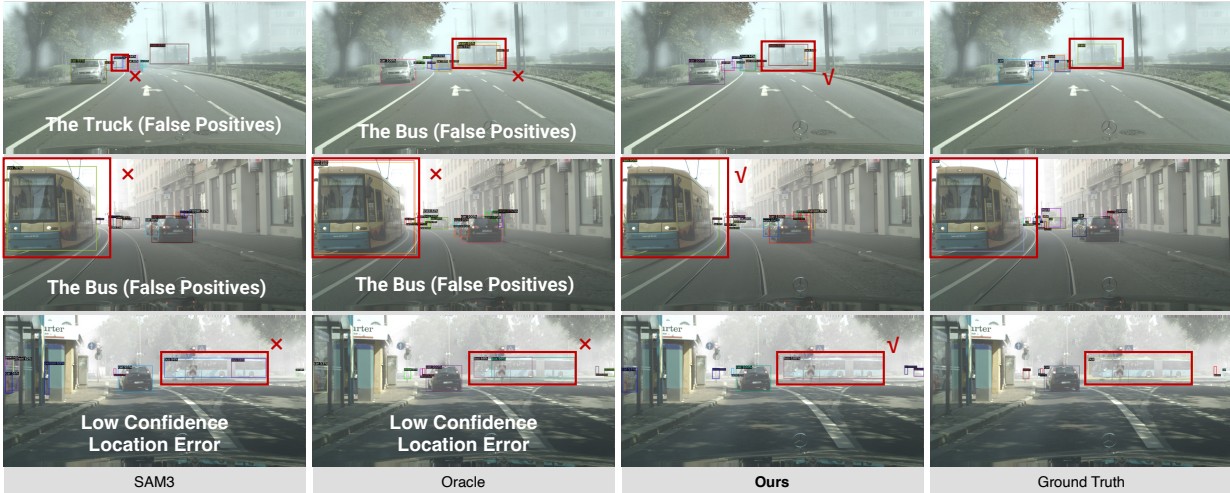

*Figure 3.* **Qualitative Comparison:** Visualizations of object detection results from SAM 3, Oracle, and the proposed method. By integrating causal refinement with domain adaptation, the proposed method refines features to better distinguish similar objects across domains, leading to robust detection and performance surpassing fully supervised baselines (Oracle).

**Ablation on Internal Component Design.** Tab. 4 (a) investigates the internal design of the SCC component within SPC, starting from the PDC-equipped R-Init baseline of 63.0% mAP. Removing specific components like Instance Normalization (IN), Attention, or Residual connection leads to varying degrees of performance drops, highlighting the necessity of each component for preserving structural integrity. The ablation results confirm the crucial role of our proposed IN and Attention operations in producing feature representations that are less sensitive to style noise and more helpful to improve semantic consistency. Further ablations of SPC are provided in Sec.B.6 & B.5.

Tab. 4 (b) explores spectral perturbation strategies within the DGCR module built upon the SPC-equipped model at 64.6% mAP. We first evaluate fixed frequency bands, which improve over the baseline by 0.3%, demonstrating the effectiveness of spectral perturbation. We then compare this against two randomized strategies: Random-Band, which randomly samples cutoff thresholds, and Random-Factor, which randomly samples the modulation factor, both within fixed ranges. The results show that both randomized causal refinement methods perform poorly. Instead, we achieved the best results **65.6%** mAP. This comparison demonstrates that our dynamic causal refinement approach, which regulates spectral perturbation intensity based on the discrepancy between teacher and student outputs on target data, outperforms both static and stochastic alternatives. Further ablations of DGCR are provided in Sec.B.3 & B.4.

### 4.4. Visualization

Fig. 3 presents a qualitative comparison of object detection outputs from SAM 3 and Oracle. The visualization highlights that while SAM 3 performs well via direct inference, it still produces false positives and object localization inac-

curacies. Our analysis reveals that after causal refinement in domain adaptation, our model significantly reduces both false positives and false negatives compared to Oracle. This outcome not only validates our approach but also provides a novel perspective for domain adaptive object detection research.

## 5. Discussion

### 5.1. Comparison with The Enhanced Oracle

To establish a rigorous and fair upper bound, we implement an Enhanced Oracle for our experiments. We contend that general enhancement techniques, which are not exclusive to unsupervised adaptation, should also be applied to the Oracle. As shown in Tab. 6, the Oracle is trained on the fully labeled target domain using the same Advanced Training Strategy detailed in Sec. B.1. This includes: (1) Modern Initialization: Consistent with our best setting, the Oracle utilizes the RegionCLIP initialization to match the backbone capacity. (2) Advanced Optimization: It employs the same Strong Augmentations and EMA mechanism used in our Robust Start phase.

Tab. 1 shows that our method unexpectedly outperforms the enhanced Oracle baseline in the C → F setting, an advantage that is not evident in other configurations. The performance gain is particularly notable for the "truck", "motorcycle", and "bus" categories. As illustrated in Fig. 3, the enhanced Oracle baseline generates false positives for the bus category, thereby compromising its performance, whereas our approach effectively mitigates this issue. On the one hand, we posit that this improvement stems from the causal refinement mechanism in our framework, which strengthens causal feature representation, enhances discrimination among similar categories, and fosters more balanced

*Table 5.* Ablation study evaluating the effect of the DGCR component when randomly discarding mid-frequency information, and when simultaneously discarding high- and low-frequency information. Note that % denotes discarding ratio.

| Frequency Information | DGCR | 0% | 20% | 40% | 60% |
|---|---|---|---|---|---|
| Mid-frequency | - | 64.55 | 48.47 | 37.94 | 31.21 |
| | ✓ | **65.64** | **45.96** | **33.39** | **27.16** |
| High- and low-frequency | - | 64.55 | 59.74 | 55.62 | 51.10 |
| | ✓ | **65.64** | **60.67** | **56.39** | **52.44** |

learning across all classes. On the other hand, we attribute this primarily to the fact that Foggy Cityscapes is synthetically derived from Cityscapes. Consequently, the domain gap is narrower and is dominated by stylistic variation rather than semantic or structural change, making adaptation more tractable. In contrast, in C → B setting where the domain distribution undergoes a more substantial shift, our method does not surpass the Oracle performance.

## 5.2. Correlation between Frequency and Causality

Prior work (Xu et al., 2023; Liu et al., 2024b) has demonstrated that domain-invariant representations contain two components: causal features and spurious factors. Causal features, which are predominantly concentrated in mid-frequency components, enhance cross-domain robustness in object detection. In contrast, spurious factors, which are primarily concentrated in high- and low- frequency bands, serve as shortcuts. Motivated by these observations (Rahaman et al., 2019) and theoretical analysis (Pinson et al., 2023), research (He et al., 2024) attributes the underlying cause of poor generalization to the fact that, during optimization, models tend to use shortcuts (Wang et al., 2022; 2023) to minimize training loss, instead of focusing primarily on learning causal factors, which determines the generalization.

To further validate the correlation between frequency and causal/spurious factors, we perform ablation experiment in the frequency information by selectively discarding either the mid-frequency band or both the high- and low-frequency bands simultaneously. As shown in the upper part of Tab. 5, discarding mid-frequency information significantly harms generalization, showing that causal factors are mainly encoded in the mid-frequency band. In contrast, as shown in the lower part of Tab. 5, when both high- and low-frequency information are selectively removed, our method (with DGCR) significantly outperforms the baseline (without DGCR). This robustness demonstrates that DGCR effectively reduces the model's reliance on spurious shortcuts present in high- and low-frequency information. Moreover, the performance of our method (with DGCR) drops more sharply when mid-frequency information is removed, further validating the effectiveness of DGCR. These results confirm that DGCR successfully shifts the model's focus away from spurious factors to causal factors.

## 5.3. Mechanism of the Causal Refinement

To improve cross-domain generalizability, we propose a dynamic perturbation method that encourages the model to focus on causal features rather than spurious factors within statistical invariance. Our approach operates within a self-distillation framework, where the distillation loss serves as a primary signal for assessing the extent to which both the teacher and student models have learned domain-invariant representations. Specifically, a high prediction discrepancy between the teacher and student models corresponds to a high distillation loss, indicating that domain-invariant representations have not been effectively learned. Conversely, when the model has successfully learned a sufficient set of shared, domain-invariant features, the distillation loss becomes relatively low. At this point, we dynamically increase the perturbation strength. This adaptive increase prevents the model from overemphasizing spurious factors within the domain-invariant representations, thereby cutting off potential shortcuts that could lead to overfitting. By doing so, the model is guided to focus on causal features throughout the entire dynamic causal refinement process.

## 6. Conclusion

In this paper, we revisit DAOD through a causal perspective, revealing that the conventional pursuit of domain invariance achieves only statistical invariance, retaining both domain-common spurious factors and causal factors. To address this, we propose the Dynamic Causal Refinement framework (DCR). First, we introduce Semantic Prediction Consistency (SPC) to enforce statistical invariance, effectively filtering out domain-specific noise. Crucially, leveraging the feedback from SPC, we devise Discrepancy-Guided Causal Refinement (DGCR). By dynamically modulating spectral perturbations on source data, DGCR selectively suppresses the dependence on domain-common spurious factors while strictly focusing on causal contents under supervision, compelling the model to focus solely on intrinsic object features. Extensive experiments validate the superior performance of DCR. By dynamically adjusting the data distribution based on the model's training state, our approach adaptively focuses on causal features for improved generalization and offers a new direction for robust representation learning.

## Acknowledgement

This work was supported by Fundamental and Interdisciplinary Disciplines Breakthrough Plan of the Ministry of Education of China (JYB2025XDXM102), the Sichuan Province Innovative Talent Funding Project for Postdoctoral Fellows (BX202405), and the Sichuan Science and Technology Program (2026NSFSC1451, 2025ZNSFSC0479). We also acknowledge the thoughtful discussions and paper revisions from Prof. Peng Wang.

## Impact Statement

This paper presents work whose goal is to advance the field of Machine Learning. There are many potential societal consequences of our work, none which we feel must be specifically highlighted here.

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

# A. Method Details

## A.1. Method Design

**Standard Supervised Training and EMA.** On the labeled source domain, the student network is optimized using the standard Faster R-CNN loss $\mathcal{L}_{sup}$, which is composed of the Region Proposal Network (RPN) loss and the ROI Head loss:

$$\mathcal{L}_{sup} = (\mathcal{L}_{rpn\_cls} + \mathcal{L}_{rpn\_reg}) + (\mathcal{L}_{roi\_cls} + \mathcal{L}_{roi\_reg}). \tag{15}$$

Simultaneously, the teacher network $\theta_{tch}$ is updated as the Exponential Moving Average (EMA) of the student $\theta_{stu}$:

$$\theta_{tch}^{(t)} = \beta_{tch}\theta_{tch}^{(t-1)} + (1 - \beta_{tch})\theta_{stu}^{(t)}. \tag{16}$$

**Semantic Prediction Consistency (SPC).** It enforces consistency on semantic context features and prediction distributions.

**Semantic Context Consistency (SCC).** We extract features from a candidate set of layers $\mathcal{P}$. For the FPN architecture, $\mathcal{P} = \{P_2, P_3, P_4, P_5\}$; for the C4 architecture, $\mathcal{P}$ corresponds to the residual blocks $\{res_2, res_3, res_4\}$. For a selected feature map $f \in \mathcal{P}$, we use Instance Normalization (IN) across the different dimensions to decouple style information:

$$\mu_c = \frac{1}{HW}\sum_{h=1}^{H}\sum_{w=1}^{W} f_{chw}, \quad \sigma_c = \sqrt{\frac{1}{HW}\sum_{h=1}^{H}\sum_{w=1}^{W}(f_{chw} - \mu_c)^2}. \tag{17}$$

**Prediction Distribution Consistency (PDC).** Before computing the consistency losses, the teacher network first performs a standard inference on the target image to generate a set of pseudo-labels based on its final predictions. Using these Pseudo-GTs, we enforce consistency through the following stages:

- RPN Consistency (Universal Anchor Sampling): In the RPN, anchors are sampled based on their IoU with gt boxes for loss computation. To ensure that the student and teacher models are optimized on the same set of anchors, we instead sample anchors using the teacher's pseudo-labels as ground truth. This produces a consistent set of indices for positive (foreground) and negative (background) anchors across both models. For these matched anchors, we extract the raw objectness logits and proposal deltas (proposal regression offsets) from both the student and teacher networks.

- ROI Consistency (Shared Proposals): To align the feature representations between the student and teacher models, we employ a shared proposal generation strategy. The student's Region Proposal Network (RPN) produced a set of region proposals, which are subsequently passed to both the student's and teacher's Region-of-Interest (RoI) heads. This ensures that both models processed identical spatial regions, eliminating variance due to proposal sampling. The teacher model then applies a confidence-based filter to these proposals, retaining only those with classification scores above a predefined threshold as foreground candidates. For this consistent set of foreground proposals, we extracted the raw classification logits and bounding-box regression deltas from the output layers of both networks.

**Loss Formulation Equivalence.** To formalize the consistency objectives between the student and teacher models, we unified the classification alignment under the theoretical framework of Kullback-Leibler (KL) Divergence. In practice, given that the teacher's parameters are frozen and its gradients detached, this formulation is mathematically equivalent to minimizing standard cross-entropy losses. This equivalence allowed for a stable and efficient implementation.

Specifically, for RPN objectness prediction, we minimized the Binary Cross-Entropy (BCE) loss between the student's predicted logits and the teacher's sigmoid-activated probabilities. For RoI head classification, we minimized the Cross-Entropy (CE) loss between the student's logits and the teacher's softmax-activated class probabilities. For all bounding-box regression tasks (both in the RPN and the RoI heads), we minimized the Smooth L1 loss between the student's and the teacher's predicted regression deltas. This loss formulation ensured a direct and gradient-stable alignment of the student's predictions with the frozen teacher's outputs across all detection sub-tasks.

**Discrepancy-Guided Causal Refinement (DGCR).** We employed the Discrete Cosine Transform (DCT) . The transformation is applied channel-wise. For a specific channel $x$ of size $H \times W$, the frequency representation $\mathcal{F}(u, v)$ is computed as:

$$\mathcal{F}(u, v) = \alpha(u)\alpha(v)\sum_{i=0}^{H-1}\sum_{j=0}^{W-1} x(i, j)\cos\left[\frac{\pi}{H}\left(i + \frac{1}{2}\right)u\right]\cos\left[\frac{\pi}{W}\left(j + \frac{1}{2}\right)v\right], \tag{18}$$

where $u \in \{0, \ldots, H-1\}$ and $v \in \{0, \ldots, W-1\}$ denote the frequency coordinates.

The normalization coefficients $\alpha(u)$ and $\alpha(v)$ are defined as:

$$\alpha(u) = \begin{cases} \sqrt{\frac{1}{H}}, & u = 0 \\ \sqrt{\frac{2}{H}}, & u \neq 0 \end{cases}, \quad \alpha(v) = \begin{cases} \sqrt{\frac{1}{W}}, & v = 0 \\ \sqrt{\frac{2}{W}}, & v \neq 0 \end{cases}. \tag{19}$$

To determine the spectral boundaries, we define the radial frequency $\rho(u,v)$ as the normalized Euclidean Distance from the DC component (origin at top-left):

$$\rho(u,v) = \sqrt{\left(\frac{u}{H}\right)^2 + \left(\frac{v}{W}\right)^2}. \tag{20}$$

Based on this design, we generate the binary masks for the static and perturbation bands.(Eq.8 & 9 in the main paper)

### A.2. Parameters and Settings

**Common Settings.** We employ Faster R-CNN as the base detector with a ResNet-50 backbone for all experiments. The backbone is initialized using two distinct pre-training strategies: standard ImageNet pre-training and RegionCLIP pre-training. For the latter, we utilize a modified ResNet-50 structure compatible with the RegionCLIP visual encoder. We conduct experiments using both the Feature Pyramid Network (FPN) and the C4 detection neck architectures. The teacher model is updated via an Exponential Moving Average (EMA) of the student's parameters with a decay rate of $\beta_{\text{tch}} = 0.9996$.

**SPC Parameters.**

- **SCC:** The configuration depends on the architecture. For FPN, we utilize features from levels $P_4$ and $P_5$. The layer-dependent temperatures are $\tau_{P4} = 1.75$ and $\tau_{P5} = 0.8$, with balancing weights $\alpha^{P4} = 0.7$ and $\alpha^{P5} = 0.3$. For C4, we use the $res4$ feature layer. The temperature is $\tau = 1.0$, weight $\alpha = 1.0$. For both architectures, the margin is $m = 0.15$ (FPN) / 0.2 (C4), and the residual scaling factor is $\lambda = 0.1$.

- **PDC:** The whole consistency loss is balanced by $\lambda_s = 1.0$ (for SCC) and $\lambda_p = 0.1$ (for PDC).

**DGCR Parameters.**

For the spectral perturbation, the Gaussian noise variance is set to $\sigma^2 = 1$. The anchor thresholds are initialized at $[\tau_1^0, \tau_2^0] = [0.005, 0.7]$ and are dynamically adjusted, subject to the constraints $\tau_1 \in [0.003, 0.03]$ and $\tau_2 \in [0.6, 0.8]$. For the feedback control mechanism, the loss EMA momentum is set to $\beta = 0.996$. The modulation parameters are configured as follows: stability constant $\xi = 10^{-6}$, sensitivity factor $\kappa = 1.5$, and amplitude limit $\eta = 0.03$.

### A.3. Experiment Details

**Datasets and Scenarios.**

- **Cross-Weather (C → F).** This scenario evaluates robustness against weather corruptions. The source domain, Cityscapes (C) (Cordts et al., 2016), consists of 2,975 training and 500 validation images of urban street scenes captured under normal weather. The target domain, Foggy Cityscapes (F) (Sakaridis et al., 2018), is a synthetic extension of Cityscapes simulated with depth-dependent fog. To provide a comprehensive assessment, we report results on two protocols in the main text: the **0.02** split (densest fog) and the **All** split (combining densities of 0.005, 0.01, and 0.02) to test extreme robustness. Note that experiments in the supplementary material are conducted exclusively on the C → F **0.02** split.

- **Cross-Camera (C → B).** This setting addresses domain shifts caused by camera intrinsics and scene structures. We use Cityscapes as the source and BDD100k (B) (Yu et al., 2020) as the target. BDD100k is a large-scale driving dataset; following prior work (Lavoie et al., 2025), we utilize the daytime subset (36,728 training and 5,258 validation images) and evaluate on the seven shared categories: person, rider, car, truck, bus, motorcycle, bicycle.

- **Synthetic-to-Real (S → C).** This measures the transferability from simulation to the real world. The source, Sim10k (S) (Johnson-Roberson et al., 2016), contains 10,000 images rendered from the GTA-V game engine. Consistent with standard conventions, we train on the 58,071 labeled car bounding boxes in Sim10k and evaluate performance exclusively on the car category of the Cityscapes validation set.

**Training Details.**

- **Data Augmentations.** We implement a hierarchical augmentation pipeline. (1) **Weak Augmentation:** Applied initially to both source and target images. This includes random horizontal flipping and multi-scale resizing. The teacher network

takes the weakly augmented target images as input. (2) **Strong Augmentation:** Applied on top of the weak augmentations exclusively for the student network input. For source images, this includes Random Color Jittering (brightness, contrast, saturation), Gaussian Blur, and Random Erasing. For target images, Random Erasing is replaced by MIC (Hoyer et al., 2023) to facilitate robust consistency learning.

- **Optimization Schedule.** Training is conducted on 8 NVIDIA A100 GPUs using the SGD optimizer (momentum 0.9, weight decay $10^{-4}$). Adhering to the "Advanced Training Strategy" (Sec. B.1), the training proceeds in two phases: (1) **Robust Start Phase:** 10k iterations of supervised training; (2) **Self-Training Phase:** 20k iterations of adaptive adaptation. The total batch size is set to 32, comprising 16 labeled and 16 unlabeled samples. The learning rate is initialized at 0.02 and warmed up over the first 1,000 iterations.

## B. Experiment Analysis

### B.1. Advanced Training Strategy

To establish a robust foundation for the self-training stage, we adopt an **Enhanced Supervised Robust Start** strategy prior to its initiation. This approach is distinguished by its simplicity and effectiveness, ensuring that the subsequent teacher-student interaction begins with high-quality guidance.

**Mechanism.** This Robust Start phase is operating as a specific supervised-only configuration of our framework. In this mode, we deactivate the unsupervised consistency branch ($\mathcal{L}_{cons}$) and the DGCR module. The student network is minimizing the standard supervised loss $\mathcal{L}_{sup}$ on the source domain using strong data augmentations. Simultaneously, the teacher network is functioning as a passive observer, updating its weights solely via Exponential Moving Average (EMA) from the student without generating supervision signals. Upon completion, the teacher's weights serve as the initialization for the subsequent self-training stage.

**Effectiveness of Components.** Tab. 6 validates the synergy of this strategy. Compared to using EMA or Strong Augmentation individually, the combined strategy (Robust Start) is achieving superior performance (e.g., a +3.9% gain over Strong-Only on ImageNet). This confirms that EMA is essential for stabilizing the learning trajectory under aggressive perturbations, thereby yielding the optimal initialization.

**Establishing a Modern and Strong Baseline.** Unlike legacy approaches that often rely on VGG backbones and basic pre-training, our framework employs a modern architecture (ResNet-50 FPN) combined with this enhanced optimization strategy. This modernization is yielding immediate performance benefits. As shown in Tab. 6, the "Robust Start" alone is achieving 49.2% and 53.6% mAP, which already surpasses the final performance of many previous methods. Building on this Robust Start phase, we conduct Standard Self-Training by enabling the teacher-student loop without our specific DCR designs. This is resulting in a remarkably strong baseline (e.g., 59.1% with ImageNet initialization and 62.3% with RegionCLIP initialization), thereby setting a rigorous standard for evaluation.

*Table 6.* **Progression of Performance.** Impact of robust strategies. "Weak" and "Strong" denote the augmentation intensity. "Robust Start" (Strong+EMA) is our chosen strategy. "Standard ST" and "Ours" denote the subsequent self-training results based on the Robust Start.

| | | Robust Start Stage | | | | Adaptation Stage | |
| --- | --- | --- | --- | --- | --- | --- | --- |
| **Task** | **Initialization** *(Weight)* | **Weak** *Only* | **Weak** *+ EMA* | **Strong** *Only* | **Robust Start** *(Strong+EMA)* | **Standard ST** *(Strong Baseline)* | **Ours (Full)** *(Final Result)* |
| Cross Domain (Ours) | ImageNet | 30.0 | 39.4 | 45.3 | **49.2** | 59.1 | **61.5** |
| | RegionCLIP | 38.5 | 43.6 | 46.4 | **53.6** | 62.3 | **65.6** |
| Full Supervised (Oracle) | RegionCLIP | - | - | - | **64.9** | - | - |

### B.2. Effect of Architecture and Initialization

**Impact of Architecture (FPN vs. C4).** Comparing the first and second rows in Tab. 7, the FPN-based architecture (61.5%) outperforms the C4-based counterpart (60.2%) by 1.3% when using the same ImageNet initialization. This indicates that the multi-scale feature pyramid provided by FPN is indeed more effective at handling object scale variations. Without the modern FPN neck or RegionCLIP initialization, the result of the C4 framework delivers a strong 60.2% mAP, demonstrating effective performance over existing state-of-the-art methods and confirming that the effectiveness of our approach stems fundamentally from the proposed causal discovery mechanism rather than being solely dependent on advanced architecture.

**Different Initialization for The Proposed Method.** In this subsection, we evaluate different network architectures and demonstrate the performance gains achieved by our proposed components. Switching from ImageNet to RegionCLIP initialization yields a substantial improvement of **4.1%** mAP (61.5% $\rightarrow$ 65.6%) as shown in Tab. 7. Crucially, this gain is larger than the intrinsic gap observed between the two initializations in the baseline setting (3.2%, from 59.1% $\rightarrow$ 62.3%). The representations from RegionCLIP provide a superior semantic foundation for our method.

*Table 7.* **Effect of Architecture and Initialization.** Experiments are conducted on the C$\rightarrow$F (0.02) split using our full framework. Performance comparison of our proposed components integrated into three baseline architectures: RegionCLIP-initialized FPN, ImageNet-initialized FPN, and ImageNet-initialized C4 backbone.

| Arch. | Init | SPC | DGCR | Person | Rider | Car | Truck | Bus | Train | Motor | Bicycle | mAP |
|-------|------|-----|------|--------|-------|-----|-------|-----|-------|-------|---------|-----|
| C4 | ImageNet | | | 60.2 | 61.2 | 74.8 | 38.1 | 56.9 | 56.9 | 49.4 | 60.4 | 57.2 |
| C4 | ImageNet | ✓ | ✓ | 61.0 | 62.1 | 75.7 | 44.8 | 63.2 | 62.3 | 52.9 | 60.0 | 60.2 |
| FPN | ImageNet | | | 64.7 | 64.6 | 76.3 | 40.4 | 61.2 | 54.2 | 50.5 | 61.0 | 59.1 |
| FPN | ImageNet | ✓ | ✓ | 64.1 | 64.8 | 77.0 | 46.6 | 66.3 | 62.0 | 51.0 | 60.1 | 61.5 |
| FPN | RegionCLIP | | | 68.9 | 71.0 | 79.0 | 45.7 | 64.7 | 48.8 | 55.6 | 64.7 | 62.3 |
| FPN | RegionCLIP | ✓ | ✓ | 69.6 | 71.8 | 79.0 | 51.3 | 71.5 | 59.6 | 59.2 | 63.0 | **65.6** |

## B.3. Discussion of Spectral Perturbation Scope Across Different Domains

In the proposed DCR framework, spectral perturbations are being applied exclusively to the source domain (labeled data). This practice is simulating potential distribution shifts and encouraging the model to learn causal-invariant features. This design raises a natural experimental question: **Should frequency perturbation also be applied to the unlabeled target domain during the adaptation phase?**

We conduct an experiment where the frequency perturbation is being applied to both the Source and Target domains (denoted as "Source + Target"). As is shown in Tab. 8, extending the spectral perturbation to the target domain is resulting in a significant performance drop, decreasing the mAP from 65.64% to 64.60%. This result is statistically indistinguishable from the baseline without any spectral perturbation (64.55%, in Tab. 4(b)).

This performance degradation arises from the inherent delicacy of unsupervised training without annotation. Perturbing the frequency components of target images disrupts the semantic consistency essential for effective self-training. Consequently, perturbations in the target domain create excessive noise, which destabilizes the learning of statistical invariance. We therefore restrict the application of DGCR strictly to the source domain.

*Table 8.* **Domain Selection of Spectral Perturbation Scope.** Comparison of DGCR application strategies: applying DGCR to the source domain only (our method) versus applying it to both the source and target domains. The "None" baseline represents the model with only SPC but no spectral perturbation.

| Method | Perturbation Domain | mAP |
|--------|---------------------|-----|
| Ours | None | 64.55 |
| | **Source Only** | **65.64** |
| | Source + Target | 64.60 ($\downarrow$ **1.04**) |

## B.4. Ablation Analysis of Guidance

In our designed framework, the Discrepancy-Guided Causal Refinement (DGCR) is operating as a **closed-loop feedback system**. Unlike static augmentation strategies, the guidance signal is actively modulating the spectral bandwidth, thereby continuously shaping the optimization trajectory throughout training. Given this inherently dynamic and cumulative influence, it is essential to ascertain that the observed performance gain is attributable to the systematic superiority of the guidance mechanism itself, rather than to a fortuitous initialization state.

To validate the robustness of our design, we report the averaged results across multiple independent runs, thereby mitigating the variance inherent in stochastic training. We compare two guidance strategies: (1) Using only the prediction discrepancy ($\mathcal{L}_{pdc}$); (2) Using the proposed consistency ($\mathcal{L}_{cons} = \mathcal{L}_{scc} + \mathcal{L}_{pdc}$).

As presented in Tab. 9, relying solely on the partial signal $\mathcal{L}_{pdc}$ is suboptimal. While it is capturing output-level misalignment, it is overlooking the mid-level feature discrepancies handled by SCC. By employing the holistic signal $\mathcal{L}_{cons}$, the model is achieving a consistent average gain of **+0.33%**. This result is confirming that the $\mathcal{L}_{cons}$ is providing a more accurate estimation of the "learning state" (defined by the alignment between source and target domains) and is demonstrating that $\mathcal{L}_{cons}$ can robustly guide the causal refinement process along a superior optimization trajectory.

*Table 9.* **Effectiveness of the Different Guidance Strategies.** ($\Delta$) indicates the relative mAP gain averaged over two independent runs, utilizing the prediction distribution consistency guidance as the baseline.

| Guidance | Component | $\Delta$ |
|---|---|---|
| Prediction Distribution Consistency | $\mathcal{L}_{pdc}$ | - |
| All | $\mathcal{L}_{scc} + \mathcal{L}_{pdc}$ | **+0.33** |

## B.5. Ablation Analysis of Prediction Distribution Consistency (PDC)

In the main paper, we introduce the Prediction Distribution Consistency (PDC) component to address the limitations of deterministic pseudo-labels. In this section, we provide a detailed ablation study to verify its effectiveness from the perspective of supervision granularity. Specifically, we analyze the impact of two critical design choices inherent in PDC: (1) The transition from deterministic label assignment to probabilistic distribution alignment; (2) The efficacy of relative offset consistency compared to coordinate regression.

We design four experimental configurations for comparison on the Cityscapes $\rightarrow$ Foggy Cityscapes (0.02) benchmark:

- **Standard Pseudo-label:** The prevailing paradigm (e.g., AT (Li et al., 2022)) where supervision is being derived by thresholding the teacher's confidence into **deterministic categorical labels**. Crucially, this approach typically ignores regression loss on the target domain.

- **Standard + Regression:** Extending the standard approach by strictly treating the teacher's pseudo-boxes as absolute Ground Truth (GT). The student is minimizing the original detection regression loss towards these coordinates.

- **PDC (Cls-only):** The employed consistency strategy that is aligning the full probabilistic category distribution between teacher and student (Eq. 6 without coordinate regression).

- **PDC (Full):** The complete strategy, incorporating both probabilistic distribution alignment and relative offset consistency on shared proposals.

*Table 10.* **Comparison of Different Pseudo-labels and Bounding Box Calculative Strategies**, "Box Offset" refers to the predicted box offset rather than box coordinates. ($\Delta$) denote the relative improvement over the standard baseline.

| Configuration | Pseudo-label | Regression Strategy | $\Delta$ |
|---|---|---|---|
| Standard Pseudo-label | Deterministic | None | - |
| Standard + Regression | Deterministic | Box Coordinate | -0.30 |
| PDC (Cls-only) | Probabilistic Dist. | None | +0.19 |
| PDC (Full) | Probabilistic Dist. | Box Offset | **+0.72** |

**Analysis of Results**. In Tab. 10, simply applying the standard approach with box regression (Standard + Regression) results in a performance drop ($\Delta = -0.30\%$). This suggests that treating the teacher's predicted boxes as ground truth is introducing noise, as the teacher's localization on the target domain inevitably contains location error.

In contrast, the strategy of Prediction Distribution Consistency (PDC) is yielding consistent improvements. Replacing deterministic labels with probabilistic distribution alignment (PDC (Cls-only)) is bringing a gain of +0.19%, indicating that richer information from the teacher model is providing more effective supervision. Furthermore, the full configuration achieves the improvement of **+0.72%**. This relative gain demonstrates that the relative box offsets are effectively mitigating the location error in pseudo-labels. It should be noted that this module accounts for only a portion of the total performance gain reported for SPC in the main paper, with the remainder being attributed to the SCC module.

### B.6. Ablation Analysis of Semantic Context Consistency

To verify whether the gain of SCC is coming from complex module design or simply feature alignment, we replace SCC with two alignment strategies: (1) **Pixel-wise L1:** Directly minimizing the $L_1$ distance between student and teacher feature maps. (2) **Global Average Pooling (GAP):** Aligning the global feature vectors after spatial pooling.

*Table 11.* **Comparison with Different Feature Alignment Strategies.** The proposed SCC is replaced with direct L1 loss and GAP-based alignment to demonstrate the necessity of the proposed design.

| Alignment Strategy | Mechanism | mAP |
|---|---|---|
| Baseline (w/o SCC) | - | 63.0 |
| Pixel-wise L1 | Direct Alignment | 63.1 |
| GAP | Global Alignment | 63.4 |
| **SCC (Ours)** | **Style-Invariant Attention** | **64.6** |

**Analysis.** As shown in Tab. 11, simple Pixel-wise L1 alignment brings negligible improvement (+0.1%), as it forces the student to mimic the teacher's noisy, domain-specific style. While GAP relaxes spatial constraints and improves performance to 63.4%, it sacrifices fine-grained semantic details. In contrast, the proposed SCC achieves a significant boost to **64.6%**. This demonstrates that effective consistency requires explicitly identifying and aligning semantic content while discarding style variations, which previous methods fail to achieve.

## C. Further Analysis

### C.1. Discussion on Adversarial Learning

Adversarial learning is excluded from the final framework owing to its significant training instability. To further validate this design and assess model stability, a multi-seed analysis is subsequently performed (Tab. 12).

The results confirm that the adversarial learning suffers from higher variance (Std. 0.37 vs. 0.23) without yielding consistent gains. Notably, while the multi-seed average of our final model (66.05%) actually exceeds the single-seed result reported in our main tables (65.64%), we retain the original reported results. We do not recompute the full suite of ablations using the higher-performing seeds or the mean to strictly avoid favorable configurations.

*Table 12.* **Ablation on Adversarial Learning.** Mean mAP and standard deviation are reported over 5 random seeds.

| Adversarial Learning | mAP | mAP (Average) ↑ | Std. ↓ |
|---|---|---|---|
| ✓ | 65.45 | 65.65 | 0.37 |
| ✗ | **65.64** | **66.05** | 0.23 |

### C.2. Parameter and Complexity Analysis

Tab. 13 presents the parameter efficiency of the proposed framework. Crucially, the proposed components are strictly confined to the training phase. Consequently, the method incurs zero additional overhead during inference, maintaining the exact same architecture and inference speed as the base detector.

*Table 13.* **Parameter Analysis.** All parameter counts are reported in Millions (M). Attn. Params are training-only components discarded during inference. RC-R50: RegionCLIP-initialized Modified ResNet-50.

| Backbone | Arch. | Base(Inference) Params | Attn. Params | Training Params |
|---|---|---|---|---|
| R-50 | C4 | 33.16 | 4.20 | 37.36 |
|  | FPN | 41.44 | 0.53 | 41.97 |
| RC-R50 | FPN | 41.46 | 0.53 | 41.99 |

**Backbone Parameters.** The RegionCLIP-initialized backbone (RC-R50) exhibits a marginal parameter increase of 0.02M compared to the standard ResNet-50 (41.46M vs. 41.44M in FPN). This slight variation stems from the structural modifications in the pre-trained backbone rather than additional functional modules. RC-R50 functions as a direct replacement for standard ResNet backbones, integrating seamlessly into the FPN detection pipeline.

**Training Efficiency.** The distillation module introduces minimal parameter overhead during training. In the FPN config-uration, the module adds only **0.53M** parameters. This is significantly more efficient than the C4 configuration (4.20M), as the module operates on the channel-reduced FPN features (256 channels) rather than the high-dimensional C4 outputs. Furthermore, compared to adversarial learning (0.59M) in **Sec. C.1**, the proposed approach achieves superior adaptation performance with smaller parameter overhead (0.53M), ensuring training stability and final performance.

*Table 14.* **Comparison with Large-scale Vision Foundation Models.** Note that these methods are evaluated without specific training, only leveraging their extensive pre-training on web-scale data. [†]: Reproduced using the official code.

| Method | Backbone | Person | Rider | Car | Truck | Bus | Train | Motor | Bicycle | mAP |
|---|---|---|---|---|---|---|---|---|---|---|
| TDA (Karmanov et al., 2024) [CVPR'24] | Swin-T | 35.3 | 4.9 | 46.4 | 22.8 | 32.3 | 0.4 | 28.7 | 29.9 | 25.1 |
| BCA (Zhou et al., 2025) [CVPR'25] | Swin-T | 41.0 | 5.6 | 47.9 | 21.9 | 33.4 | 0.9 | 28.5 | 29.8 | 26.1 |
| GDINO (Liu et al., 2024a) [ECCV'24] | Swin-T | 30.1 | 3.4 | 46.3 | 22.4 | 32.0 | 0.1 | 28.0 | 28.9 | 23.9 |
| TDA (Karmanov et al., 2024) [CVPR'24] | Swin-B | 38.3 | 25.5 | 50.1 | 30.5 | 45.9 | 12.0 | 33.1 | 39.1 | 34.3 |
| BCA (Zhou et al., 2025) [CVPR'25] | Swin-B | 36.5 | 24.4 | 50.8 | 30.8 | 45.3 | 20.3 | 33.2 | 42.5 | 35.5 |
| GDINO (Liu et al., 2024a) [ECCV'24] | Swin-B | 35.0 | 20.4 | 51.6 | 30.0 | 45.7 | 0.9 | 35.2 | 32.1 | 31.3 |
| SAM 3[†] (Carion et al., 2025) | - | 55.9 | 61.3 | 75.9 | 49.3 | 64.4 | **60.2** | 53.1 | 56.0 | 59.5 |
| **Ours** | ResNet-50 | **69.6** | **71.8** | **79.0** | **51.3** | **71.5** | 59.6 | **59.2** | **63.0** | **65.6** |

## C.3. Comparison with Large-scale Vision Foundation Models

The emergence of large-scale vision foundation models trained on web-scale data (e.g., O365, GoldG, SA-1B) raises a fundamental question: **Is domain adaptation still necessary if a model has "seen it all"? We argue that the answer remains affirmative, at least in the context of autonomous driving.** We compare our method with existing approaches based on Grounding DINO (Liu et al., 2024a) (In Tab. 14). The results demonstrate that our proposed approach significantly outperforms detection methods built upon vision foundation models (VFMs), reinforcing the necessity of in-depth domain adaptation research for autonomous driving scenarios. Meanwhile, we further compare our method with the recent segmentation foundation model SAM 3 (Carion et al., 2025). Although the SAM 3 achieves competitive results, our method maintains superior inference speed, effectively meeting the real-time requirements of typical deployment environments.

*Table 15.* **Inference Speed Comparison.** The inference latency (milliseconds per image) and throughput (Frames Per Second) are evaluated on the single A100 GPU. "Rel." denotes the speed relative to the proposed method.

| Model | Latency (ms) | FPS | Rel. Speed |
|---|---|---|---|
| Grounding DINO-T | 178.9 | 5.6 | 0.23× |
| Grounding DINO-B | 229.7 | 4.4 | 0.18× |
| SAM 3 | 664.6 | 1.5 | 0.06× |
| **Ours (Faster R-CNN)** | **40.7** | **24.6** | **1.00×** |

