# OpenReview forum: "Domain Adaptive Object Detection via Dynamic Causal Refinement"
_ICML.cc/2026/Conference — ICML 2026 regular_

### Official Review · Reviewer_b5Sc · 2026-03-11

**Soundness:** 3
**Presentation:** 3
**Significance:** 3
**Originality:** 3
**Overall Recommendation:** 4
**Confidence:** 4

**Summary:**

This paper focus on domain adaptive object detection, aiming to transfer the labeled source domain model to the unlabeled target domain. Existing methods mainly pursue statistical invariance rather than causal invariance. Therefore, a dynamic causal refinement framework is proposed, which consists of two modules. SPC removes style information, uses cross-attention mechanism to align the intermediate features of the teacher and the student, utilizes KL divergence to align the complete category probability distribution, and introduces detection box regression consistency constraint; DGCR uses the teacher-student network to predict differences and dynamically adjust the spectral perturbation intensity.

**Compliance With Llm Reviewing Policy:**

Affirmed.

**Final Justification:**

Thank you for the detailed rebuttal. My main concerns have been addressed, and I raise my score accordingly.

**Key Questions For Authors:**

Why did the results of this method even surpass those of the fully supervised Oracle in Table 1?

In the paper, the initial threshold of DGCR is set to [0.005, 0.7], with the constraint ranges being [0.003, 0.03] and [0.6, 0.8]. Are these values universally applicable across different datasets?

**Limitations:**

yes

**Strengths And Weaknesses:**

Strengths:


1. The article first analyzed the difficulties of domain adaptive object detection from a causal perspective, decomposing statistical invariance into causal factors and false factors, and revealing that the false factors are the main cause of performance degradation.

2. The authors propose DGCR module that  dynamically adjusts the intensity of spectral perturbation through teacher-student prediction based on differences, allowing the model to balance between learning statistical invariance and causal invariance.

3. Extensive experiments were conducted to prove the effectiveness of this method.

Weaknesses:

1 .Why do intermediate frequencies represent causal factors, while high and low frequencies represent spurious factors? This should be explained.

2. Is there a correlation between the differences in teacher-student network predictions and the causal characteristics? The differences in predictions might arise from false label noise or domain shift.

3. The DGCR method is fundamentally based on the structural properties of images and cannot be effectively extended to other modalities. When considering 3D object detection, point clouds are irregular and performing frequency domain decomposition would result in the loss of semantic information.

4. This method first performs the DCT operation on the image to obtain the spectrum, then perturbs certain frequency bands, and finally uses it for training. What's the difference between this operation and data augmentation?

---

> ### Author Rebuttal · Authors · 2026-03-31
>
> Dear reviewer b5Sc, thank you for your professional and insightful comments on our paper. Your feedback is extremely valuable and plays a crucial role in improving the quality of our work. In the following, we address your questions one by one. We sincerely hope this response helps resolve your concerns.
>
> **Q1**: Explanation of the link between causality and frequency.
>
> **Empirical Explanation**: **Prior work[1, 2] has shown that domain-invariant representations consist of two parts: causal features, concentrated in mid-frequency components, which enhance cross-domain robustness in object detection;  and spurious factors, prevalent in low- and high-frequency bands, which serve as shortcuts for minimizing loss.** Motivated by these observations[3] and theoretical analysis[4], research[5] attributes the underlying cause of poor generalization to the fact that, during optimization, models tend to latch onto these spurious factors as shortcuts[6,7] to minimize training loss.
>
> **Experimental Explanation**: To further validate the connection between frequency-domain information and causal/spurious factors, and to demonstrate that our model focuses more on causal features, we conducted the following analyses. We apply random dropout in the frequency domain. **Our method (w/ DGCR) outperforms the baseline (w/o DGCR) when low- and high- frequency information is removed**, as shown in the following table. This robustness demonstrates that DGCR shifts the model's focus to learn causal features from the mid- frequency band, reducing its reliance on spurious shortcuts. Furthermore, **the greater performance drop in our method (w/ DGCR) when mid- frequency information is removed**, which clearly demonstrates the efficacy of our DGCR, as it successfully shifts the model's focus from spurious factors to the causal features.
>
> **Table**: Evaluation under random frequency information dropout.
>
> |Drop Frequency Information|DGCR|0%|20%|40%|60%|
> |:-|:-:|:-|:-|:-|:-|
> |Mid- Frequency|-|64.55|48.47|37.94|31.21|
> |Mid- Frequency|✓|65.64|45.96|33.39|27.16|
> |Low- and high- Frequency|-|64.55|59.74|55.62|51.10|
> |Low- and high- Frequency|✓|65.64|60.67|56.39|52.44|
>
> **Q2**: The correlation between the differences in teacher-student network predictions and the causal feature.
>
> High prediction differences between teacher and student models typically correspond to high loss, indicating that domain-invariant representations have not yet been effectively learned.
> Once the model has learned a sufficient set of shared, domain-invariant features, indicated by a relatively low loss, we dynamically increase the perturbation strength to prevent the model from overemphasizing spurious factors within domain-invariant features, as shown in Fig.1(Line 10), thereby cutting off its potential shortcut to overfit and guiding the model to focus on causal features throughout the dynamic causal refinement process.
>
> **Q3**: The challenge of extending DGCR to point clouds
>
> Point cloud analysis and image processing are distinct tasks, typically addressed by independent, non-transferable research methods. For irregular data like point clouds, the analogous approach would be to project the data into a different latent space. In this space, stable geometric structures (the invariant signal) can be conceptually separated from sensor-specific noise or density variations (the variant noise). Therefore, we view the extension to 3D detection not as a limitation, but as a novel direction for future work.
>
> **Q4**:Difference between DGCR and data augmentation?
>
> Inspired by causal theory[1], we design a novel, closed-loop augmentation framework specifically designed for DAOD, which advances beyond conventional augmentation by dynamically generating domain shifts through perturbations in both low- and high-frequency. The model can access different data distributions in training and engages in continuous causal refinement, progressively raising the generalization upper bound.
>
> **Q5**: The superiority over the fully supervised oracle.
>
> Previous work[8] has also reported similar superiority over the fully supervised oracle. In **Supplementary B.2 (Line 762)**, we have also analyzed the preliminary reason. We attribute this primarily to the fact that Foggy Cityscapes is synthetically derived from Cityscapes. Consequently, the domain gap is narrower and is dominated by stylistic variation rather than semantic or structural change, making adaptation eaiser. In contrast, in other settings where the domain distribution undergoes a more substantial shift, our method does not surpass the oracle performance.
>
> **Q6**: Consistency of DGCR's threshold parameters across datasets.
>
> We employed a fixed parameter set uniformly across all experiments without dataset-specific optimization.
>
> **Reference**
>
> [8]Cross-Domain Adaptive Teacher for Object Detection. CVPR, 2022.
>
> **I apologize for the space constraints here. Regarding references[1-7], please refer to my responses to Reviewer 7Twa's Q1.**

---

> > ### Author Rebuttal · Reviewer_b5Sc · 2026-04-06
> >
> > I would like to express my sincere gratitude to the author for the efforts made to address my questions. Regarding issues such as the correlation between teacher-student prediction differences and causal characteristics, the extension of DGCR to other modalities, and the differences between DGCR and data augmentation, all of my questions have been fully answered. However, regarding the reason why the performance exceeds the fully supervised benchmark model, the author did not provide a clear response or empirical evidence. There is still a lack of a clear and reasonable explanation in this regard. I can adjust the overall score accordingly.

---

> > > ### Author Response · Authors · 2026-04-07
> > >
> > > Dear Reviewer b5Sc,
> > >
> > > Thank you for your acknowledgment of our response. We are very happy to have addressed your other concerns and sincerely appreciate your recognition of our work. We sincerely apologize for the lack of clarity about why the performance exceeds the fully supervised benchmark model in the Rebuttal stage.
> > >
> > > **Result Analysis**: As shown in the following table, **our method does not surpass the Oracle in all categories**; We observe that the improvement in Average Precision (AP) is primarily concentrated in the limited categories: **Truck**, **Bus**, and **Motor**. Among these, the **Bus** category shows the most significant gain. Because mAP is an average across all categories, the substantial gains in **Bus** outweigh minor decreases in performance in other categories, resulting in a higher mAP than the Oracle.
> > >
> > > **Table**: Comparison between Ours and Oracle.
> > >
> > > |Method|Person|Rider|Car|**Truck**|**Bus**|Train|**Motor**|Bicycle|mAP|
> > > |:---:|:---:|:---:|:---:|:---:|:---:|:---:|:---:|:---:|:---:|
> > > |Ours |69.6|71.8|79.0|**51.3**|**71.5**|59.6|**59.2**|63.0|65.6|
> > > |Oracle|74.6|70.9|82.8|48.3|62.4|60.7|55.7|64.1|64.9|
> > >
> > >
> > > **Visualization Explanation in Bus**: As visualized in **the manuscript (Line 400)**, the Oracle exhibits misclassifications of **Train** as **Bus** (False Positives). Since precision ($P$) is calculated as  $P = \frac{TP}{TP + FP}$, an increase in false positives ($FP$) directly decreases precision  ($P$). **In contrast, by focusing on causal features, our method successfully suppresses the spurious similarity between categories that leads to misclassification, thereby alleviating false positives, as described in Supplementary B.2 (Line 76)**.
> > >
> > > **Dataset Explanation**: As shown in the following table, we compared Source-only and Oracle experiments, and the results indicate that C$\to$F has a smaller domain gap than C$\to$B settings. This is because the Foggy Cityscapes dataset is synthetically derived from Cityscapes, resulting in a narrower domain gap dominated by **stylistic variation** rather than semantic or structural change. This simplifies the adaptation process. **Empirical evidence from previous studies [1-5] also confirms that DA methods can achieve stronger performance than the Oracle in the C$\to$F setting.**
> > >
> > > **Table**: Comparison between Source-only and Oracle.
> > >
> > > |Setting| Source-only | Oracle | Gap |
> > > | :---: | :---: | :---: | :---: |
> > > | C$\to$F | 53.6 | 64.9 | **-11.3** |
> > > | C$\to$B | 38.9 | 64.8 | -25.9 |
> > >
> > > We are ready to provide further explanations or revisions as requested, and respectfully hope that you could reconsider your recommendation.
> > >
> > > Thank you again for your time and constructive feedback.
> > >
> > > Sincerely,
> > >
> > > Authors of Submission 938
> > >
> > > [1]Cross-domain Adaptive Teacher for Object Detection. CVPR, 2022.
> > >
> > > [2]Learning Domain Adaptive Object Detection with Probabilistic Teacher. ICML, 2022.
> > >
> > > [3]Contrastive Mean Teacher for Domain Adaptive Object Detectors. CVPR, 2023.
> > >
> > > [4]Cat: Exploiting Inter-class Dynamics for Domain Adaptive Object Detection. CVPR, 2024.
> > >
> > > [5]Align and Distill: Unifying and Improving Domain Adaptive Object Detection. TMLR, 2025.

---

### Official Review · Reviewer_VtAn · 2026-03-11

**Soundness:** 2
**Presentation:** 3
**Significance:** 3
**Originality:** 3
**Overall Recommendation:** 4
**Confidence:** 3

**Summary:**

This paper addresses Domain Adaptive Object Detection (DAOD) by challenging the traditional pursuit of statistical invariance, arguing that it often inadvertently retains domain-common spurious factors. The authors propose a Dynamic Causal Refinement (DCR) framework comprising Semantic Prediction Consistency (SPC) and Discrepancy-Guided Causal Refinement (DGCR). Specifically, DGCR applies dynamic frequency-domain perturbations guided by the discrepancy signal from SPC to progressively suppress spurious factors and highlight causal representations. Extensive experiments on standard benchmarks demonstrate that the proposed method significantly outperforms current state-of-the-art counterparts

**Compliance With Llm Reviewing Policy:**

Affirmed.

**Final Justification:**

Thank you for the detailed rebuttal. My main concerns have been addressed, and I will raise my score accordingly.

**Key Questions For Authors:**

My main questions and concerns revolve around the claimed "causal invariant factors," which currently lack rigorous theoretical or experimental justification. While the initial motivation is solid, I am afraid the core claims are not well-supported by the current model design, theoretical derivations, or experimental evaluations (as detailed in the Weaknesses section). The authors are recommended to provide direct evidence for the causal / shortcut-suppression story.

**Limitations:**

Overall, I see a promising and empirically strong DAOD paper, but not yet a fully convincing causal-learning paper. The proposed methodology is undoubtedly useful and effective, leading to strong SOTA results. However, the causal claims seem substantially overstated relative to the provided theoretical and empirical evidence. Addressing the alignment between the paper's narrative and its actual mechanisms will be crucial for the rationality of the paper.

**Strengths And Weaknesses:**

**Strengths**
1. Well-motivated and insightful perspective: The paper identifies a critical limitation in existing DAOD methods—the persistent entanglement of true causal factors with domain-common spurious factors within standard statistical invariance. The conceptual disentanglement of these factors is intuitive and tackles a core bottleneck in domain adaptation.
2. Empirically strong and comprehensive evaluation: The authors provide extensive experiments in the main paper and the appendix. The proposed method achieves significant performance improvements, establishing a new state-of-the-art on several benchmarks.
3. Clear presentation: The paper is well-written, logically organized, and highly readable. Their writing and paper style effectively convey the high-level intuition and the technical pipeline.

**Weaknesses**
1. Misalignment between the "causal" claims and the methodological design: The paper's core narrative relies heavily on causal terminology, yet the methodology is fundamentally based on frequency-domain filtering. The approach hinges on a strong assumption that causal factors reside strictly in mid-frequency bands, while high and low frequencies dictate spurious factors. However, they are inherently statistical properties of the data distribution rather than true causal factors. Using frequency perturbations as a proxy for causal intervention makes the "causal learning" claim feel substantially overstated and somewhat rigid.
2. Lack of direct evidence for causal feature extraction: While the paper demonstrates impressive improvements in mAP, it lacks targeted theoretical proofs or experiments validating that the model actually isolates true causal factors and suppresses spurious ones. Standard performance metrics and qualitative visualizations prove enhanced robustness, but they do not directly support the underlying causal narrative.
3. Disproportionate reliance on initialization: The ablation study reveals that a substantial portion of the performance gain stems from the robust initialization provided by RegionCLIP. For instance, Table 3 shows that transitioning from ImageNet to RegionCLIP alone provides a major +3.2\% mAP boost. This gain rivals the cumulative improvements introduced by the proposed SPC and DGCR modules (+3.3\% mAP to reach 65.6\%), which somewhat dilutes the perceived contribution of the paper's core innovations.

---

> ### Author Rebuttal · Authors · 2026-03-31
>
> Dear reviewer VtAn, thank you for your professional and insightful comments on our paper. Your feedback is extremely valuable and plays a crucial role in improving the quality of our work. In the following, we address your questions one by one. We sincerely hope this response helps resolve your concerns.
>
> **Q1**: Lack of direct evidence for causal feature extraction.
>
> **Empirical Evidence**: **Prior work[1, 2] has shown that domain-invariant representations consist of two parts: causal features, concentrated in mid-frequency components, which enhance cross-domain robustness in object detection;  and spurious factors (non-causal features), prevalent in low- and high-frequency bands, which serve as shortcuts for minimizing loss.** Motivated by these observations[3] and theoretical analysis[4], research[5] attributes the underlying cause of poor generalization to the fact that, during optimization, models tend to latch onto these spurious factors as shortcuts[6-7] to minimize training loss.
>
> **Experimental Evidence**: To further validate the connection between frequency-domain information and causal/spurious factors, and to demonstrate that our model focuses more on causal features, we conducted the following analyses. We apply random dropout in the frequency domain. **Our method (w/ DGCR) outperforms the baseline (w/o DGCR) when low- and high- frequency information is removed**, as shown in the following table. This robustness demonstrates that DGCR shifts the model's focus to learn causal features from the mid- frequency band, reducing its reliance on spurious shortcuts. Furthermore, **the greater performance drop in our method (w/ DGCR) when mid- frequency information is removed**, which clearly demonstrates the efficacy of our DGCR, as it successfully shifts the model's focus from spurious factors(low- and high- frequency) to the causal features(mid- frequency).
>
> **Table**: Evaluation under random frequency information dropout.
>
> |Drop Frequency Information|DGCR|0%|20%|40%|60%|
> |:---|:---:|:---|:---|:---|:---|
> |Mid- Frequency|-|64.55|48.47|37.94|31.21|
> |Mid- Frequency|✓|65.64|45.96|33.39|27.16|
> |Low- and high- Frequency|-|64.55|59.74|55.62|51.10|
> |Low- and high- Frequency|✓|65.64|60.67|56.39|52.44|
>
>
> **Q2**: Misalignment between the causal claims and the methodological design.
>
> **Prior works[1, 2] had linked frequency analysis to causality in cross-domain generalization and used frequency perturbations as a proxy for causal intervention**. Inspired by them[1, 2], we design an adaptive, closed-loop perturbation mechanism for causal feature refinement within domain-invariant representations. Specifically, once the model has learned a sufficient set of shared, domain-invariant features, indicated by a relatively low loss, we dynamically increase the perturbation strength to prevent the model from overemphasizing spurious factors within domain-invariant features, as shown in Fig.1(Line 10), thereby cutting off its potential shortcut to overfit and guiding the model to focus on causal features throughout the dynamic causal refinement process.
>
>
> **Q3**: Disproportionate reliance on initialization.
>
> To isolate the contribution of our architecture from the initialization, we conducted dedicated ablation studies with different pre-trained weights **(see Supplementary B.3, Table 6)(Line 798)**. The results clearly show that our method provides consistent improvement with ImageNet weights. Crucially, when using the same ImageNet initialization as baseline models, our approach still achieves SOTA performance(61.5\%). **This demonstrates that the performance gains are a direct consequence of our novel SPC and DGCR modules. Moreover, this compatibility demonstrates that initializing the model with other standard pre-trained weights not only produces conflict with our approach but also provides a stable foundation for further enhancement.**
>
> **Reference**
>
> [1]Multi-view Adversarial Discriminator: Mine the Non-causal features for Object Detection in Unseen Domains. CVPR, 2023.
>
> [2]Unbiased Faster R-CNN for Single-source Domain Generalized Object Detection. CVPR, 2024.
>
> [3]On the Spectral Bias of Neural Networks. ICML, 2019.
>
> [4]Linear CNNs Discover the Statistical Structure of the Dataset Using Only the Most Dominant Frequencies. ICML, 2023.
>
> [5]Towards Combating Frequency Simplicity-biased Learning for Domain Generalization. NeurIPS, 2024.
>
> [6]Frequency Shortcut Learning in Neural Networks. NeurIPSW, 2022.
>
> [7]What do Neural Networks Learn in Image Classification? A Frequency Chortcut Perspective. ICCV, 2023.

---

> > ### Author Rebuttal · Reviewer_VtAn · 2026-04-03
> >
> > All my concerns have been addressed.

---

> > > ### Author Response · Authors · 2026-04-03
> > >
> > > Dear Reviewer VtAn,
> > >
> > > Thank you for your acknowledgment of our response.  We are very happy to have addressed all your concerns and sincerely appreciate your recognition of our work.
> > >
> > > Based on the clarifications and additional evidence provided in our response, we respectfully hope that you could reconsider your recommendation.
> > >
> > > Thank you again for your time and constructive feedback.
> > >
> > > Sincerely,
> > >
> > > Authors of Submission 938

---

### Official Review · Reviewer_CrM2 · 2026-03-12

**Soundness:** 3
**Presentation:** 3
**Significance:** 3
**Originality:** 3
**Overall Recommendation:** 4
**Confidence:** 5

**Summary:**

This paper addresses the Domain Adaptive Object Detection (DAOD) problem by arguing that exisiting DAOD methods overly focus on statisitcal invariance but fail to distinguish between truely causal object cues and domain-common cues such as shared enviroments. Authors propose DCR, which contains two main components: Semantic Prediction Consistency (SPC) enforces student-teacher semantic feature consistency and predictiction-distribution consistency to filter domain-specific cues; and Discrepancy-Guided Causal Refinement (DGCR) which uses the student-teacher consistency loss as a feedback to dynamically modulate the frequency-space perturbations on source images during training. Experiments show strong improvements over SOTA as well as outperforming oracle baselines.

**Compliance With Llm Reviewing Policy:**

Affirmed.

**Key Questions For Authors:**

Is there evidence that supports that DGCR focuses causal object information, rather than simply acting as a adaptive regulariser?
How sensitive is the method to its hyperparameters?

How much of the gain comes from SPC alone versus the discrepancy-guided coupling between SPC and DGCR?

How well does the method generalize across detector architectures?

The gap between dynamic and fixed-band perturbations is 0.7 on Cityscapes to Cityscapes Foggy, is this gap consistant on other benchmarks?

**Limitations:**

Yes

**Strengths And Weaknesses:**

The paper is well presented and clearly explains each component of their method.
The experimentations and ablation studies are remarkably through. The comparision to an 'improved' oracle is also transparent and shows that performance gains come from the authors contributions rather than training 'tricks'.
Performance on benchmarks are also significantly stronger than prior SOTA methods.


There is currently no formal causal model and the link between mid-frequency bands and causal content relies on empirical observations rather than direct validation.
The ablation studies indicate taht SPC does most of the heavy lifting compared to DGCR. And a simple fixed-band perturbation already research 0.7 low than the full dynamic version and is much simpler.
Currently all experiments use FRCNN and ResNet50. It is not shown how general this method currently is.
The methods introduce many hyperparameters across both modules. A sensitivity analysis of these hyperparameters would be useful to show whether per-benchmark tuning is require or if it would be truely self-regulating.

---

> ### Author Rebuttal · Authors · 2026-03-31
>
> Dear reviewer CrM2, thank you for your professional and insightful comments on our paper. Your feedback is extremely valuable and plays a crucial role in improving the quality of our work. In the following, we address your questions one by one. We sincerely hope this response helps resolve your concerns.
>
> **Q1**: Evidence supports that DGCR focuses on causal object information.
>
> **We adopt the view from prior work[1] that domain-invariant representations consist of two parts: causal features, concentrated in mid-frequency components, which enhance cross-domain robustness in object detection;  and spurious factors (non-causal features), prevalent in low- and high-frequency bands, which serve as shortcuts for minimizing loss.**
>
> **Experimental Evidence**: To further validate the connection between frequency-domain information and causal/spurious factors, and to demonstrate that our model focuses more on causal features, we conducted the following analyses. We apply random dropout in the frequency domain. **Our method (w/ DGCR) outperforms the baseline (w/o DGCR) when low- and high- frequency information is removed**, as shown in the following table. This robustness demonstrates that DGCR shifts the model's focus to learn causal features from the mid- frequency band, reducing its reliance on spurious shortcuts. Furthermore, **the greater performance drop in our method (w/ DGCR) when mid- frequency information is removed**, which clearly demonstrates the efficacy of our DGCR, as it successfully shifts the model's focus from spurious factors(low- and high- frequency) to the causal features(mid- frequency).
>
> **Table**: Evaluation under random frequency information dropout.
>
> |Drop Frequency Information|DGCR|0%|20%|40%|60%|
> |:-|:-:|:-|:-|:-|:-|
> |Mid- Frequency|-|64.55|48.47|37.94|31.21|
> |Mid- Frequency|✓|65.64|45.96|33.39|27.16|
> |Low- and high- Frequency|-|64.55|59.74|55.62|51.10|
> |Low- and high- Frequency|✓|65.64|60.67|56.39|52.44|
>
> **Q2**: Gain comes from SPC alone versus coupling between SPC and DGCR.
>
> As reported in **Tab.3 (Line 351)** integrating both SPC and DGCR leads to a 3.3\% improvement over the baseline. Applied separately, SPC brings a 1.7\% gain, while DGCR delivers a similarly sized improvement. This coordinated design enables the two modules to complement each other, resulting in superior overall performance.
>
> **Q3**: Generalization across detector architectures.
>
> We conducted preparatory experiments with the DGCR module in a YOLOv5m-based network. As shown in the following table, integrating DGCR yields a clear performance gain compared to the baseline without it. It is worth mentioning that our method is fundamentally architecture-agnostic: SPC operates on feature representations and final detection outputs, while DGCR operates on the input image spectrum. Neither module is intrinsically tied to a specific network architecture.
>
> **Table**: Performance across detector architectures.
>
> |Architecture|mAP|
> |:-|:-:|
> |DA-YOLO(Baseline)|56.95|
> |DA-YOLO(w/DGCR)|59.16|
>
> **Q4**: Performance across other benchmarks, comparing dynamic versus fixed-band perturbations.
>
> As shown in the following table, the performance advantage of our dynamic perturbation method over the fixed-band baseline is across other benchmarks. In addition, the 0.7\% gap represents the optimal result for a **carefully tuned** fixed band on the C$\to$F benchmark. In contrast, our method does not require careful fine-tuning of frequency allocation, as this process is fully adaptive.
>
> **Table**: Performance comparison between Fixed-Band and our DGCR across different benchmarks.
>
> |Benchmark|Fixed Band|w/DGCR|Gain|
> |:-:|:-:|:-:|:-:|
> |C$\to$F|64.88|65.64|+0.76|
> |C$\to$B|51.93|52.77|+0.84|
> |S$\to$C|73.00|73.91|+0.91|
>
> **Q5**: Hyperparameter sensitivity.
>
> We clarify the parameters in Supplementary Material A.2: all parameters for the two modules are listed, but only some are tunable hyperparameters. The PDC loss coefficient in the SPC was set solely to balance loss magnitudes. Only $\kappa$ and $\eta$ are important hyperparameters: $\kappa$ is for regulating the guidance signal, and $\eta$ is for regulating the range of perturbation. Others are inherited from existing studies.
>
> In the following table, $\kappa$ exhibits relatively stable characteristics(with 0.5 even yielding better results), while $\eta$ shows strong sensitivity. This is because $\eta$ controls the coefficient of the frequency band disturbance range. If too large, it disrupts the learning of domain-invariant features. If too small, it becomes ineffective, similar to a fixed-band perturbation.
>
> **Table**: Ablation study on $\kappa$.
>
> |$\kappa$|0.5|1.2|1.5|1.8|
> |:-:|:-:|:-:|:-:|-:|
> |Ours|**65.94**|65.31|*(65.64)*|64.91|
>
> **Table**: Ablation study on $\eta$.
>
> |$\eta$|0.02|0.03|0.04|
> |:-:|:-:|:-:|:-:|
> |Ours|65.08|*(65.64)*|64.77|
>
> **Reference**
>
> [1]Multi-view Adversarial Discriminator: Mine the Non-causal features for Object Detection in Unseen Domains. CVPR, 2023.

---

> > ### Author Rebuttal · Reviewer_CrM2 · 2026-04-03
> >
> > The authors have addressed my concerns. The results of the paper are impressive. Would  the authors plan to release the open source code on acceptance for reproducibility?

---

> > > ### Author Response · Authors · 2026-04-03
> > >
> > > Dear Reviewer CrM2,
> > >
> > > Thank you for your acknowledgment of our response. We are very happy to have addressed all your concerns and sincerely appreciate your recognition of our work. We will fully release our code for reproducibility upon acceptance of this work.
> > >
> > > Thank you again for your time and constructive feedback.
> > >
> > > Sincerely,
> > >
> > > Authors of Submission 938

---

### Official Review · Reviewer_7Twa · 2026-03-13

**Soundness:** 3
**Presentation:** 3
**Significance:** 3
**Originality:** 3
**Overall Recommendation:** 4
**Confidence:** 3

**Summary:**

This paper studies unsupervised domain adaptive object detection and argues that existing methods focusing on statistical invariance may still preserve domain-shared spurious factors. To address this, the paper proposes dynamic causal refinement, which combines teacher-student consistency learning with a discrepancy-guided frequency refinement module. The method includes semantic prediction consistency for feature and prediction alignment, and discrepancy-guided causal refinement for dynamically adjusting frequency-band perturbations based on teacher-student discrepancy. Experiments on standard DAOD benchmarks show strong improvements over prior methods.

**Compliance With Llm Reviewing Policy:**

Affirmed.

**Final Justification:**

After the detailed rebuttal, I keep my original rating.

**Key Questions For Authors:**

- The current justification is plausible, but it remains largely intuitive. A clearer empirical or theoretical explanation would help strengthen the central claim of the paper.
- The paper includes ablations, but it would be helpful to further clarify whether the gains mainly come from making the perturbation adaptive, rather than from the broader causal motivation.
- The method is implemented through discrepancy-guided frequency refinement, rather than explicit causal modeling. A more precise discussion of what is meant by causal refinement would help readers assess the contribution more accurately.

**Limitations:**

No. The paper would benefit from a more explicit discussion of its main limitations, especially the fact that the causal interpretation is only indirectly supported and that the link between frequency components and causal/spurious factors remains largely empirical.

**Strengths And Weaknesses:**

Strengths
- The paper is built around a relevant motivation. It points out that statistical alignment in DAOD does not necessarily prevent the detector from relying on domain-shared contextual cues, which is a reasonable starting point for the method.
- The dynamic adjustment mechanism is a meaningful design choice. Using teacher-student discrepancy to control the refinement process is more adaptive than using a fixed perturbation strategy.
- The experimental section covers several standard DAOD benchmarks and includes ablations on the main components. This gives reasonable support to the claim that both parts of the framework contribute to the reported performance.

Weaknesses
- While the paper is framed around causal and spurious factors, the implemented method is still fundamentally a teacher-student framework with discrepancy-guided frequency perturbation. The work is better understood as a causal-motivated refinement strategy than a genuinely causal DAOD method.
- Frequency-domain perturbation and teacher-student consistency are both familiar ingredients. The most novel part is the dynamic closed-loop adjustment, which is useful, but the overall framework is still an extension of existing design patterns rather than a fundamentally new paradigm.
- The paper relies on the intuition that certain frequency components are more associated with domain-specific shortcut information, but this connection remains largely empirical and is not deeply justified.

---

> ### Author Rebuttal · Authors · 2026-03-31
>
> Dear reviewer 7Twa, thank you for your professional and insightful comments on our paper. Your feedback is extremely valuable and plays a crucial role in improving the quality of our work. In the following, we address your questions one by one. We sincerely hope this response helps resolve your concerns.
>
> **Q1**: Explanation of the link between causality, frequency, and shortcuts.
>
> **Empirical Explanation**: **Prior work[1, 2] has shown that domain-invariant representations consist of two parts: causal features, concentrated in mid-frequency components, which enhance cross-domain robustness in object detection;  and spurious factors (non-causal features), prevalent in low- and high-frequency bands, which serve as shortcuts for minimizing loss.** Motivated by these observations[3] and theoretical analysis[4], research[5] attributes the underlying cause of poor generalization to the fact that, during optimization, models tend to latch onto these spurious factors as shortcuts[6, 7] to minimize training loss.
>
> **Experimental Explanation**: To further validate the connection between frequency-domain information and causal/spurious factors, and to demonstrate that our model focuses more on causal features, we conducted the following analyses. We apply random dropout in the frequency domain. **Our method (w/ DGCR) outperforms the baseline (w/o DGCR) when low- and high- frequency information is removed**, as shown in the following table. This robustness demonstrates that DGCR shifts the model's focus to learn causal features from the mid- frequency band, reducing its reliance on spurious shortcuts. Furthermore, **the greater performance drop in our method (w/ DGCR) when mid- frequency information is removed**, which clearly demonstrates the efficacy of our DGCR, as it successfully shifts the model's focus from spurious factors(low- and high- frequency) to the causal features(mid- frequency).
>
> **Table**: Evaluation under random frequency information dropout.
>
> |Drop Frequency Information|DGCR|0%|20%|40%|60%|
> |:---|:---:|:---|:---|:---|:---|
> |Mid- Frequency|-|64.55|48.47|37.94|31.21|
> |Mid- Frequency|✓|65.64|45.96|33.39|27.16|
> |Low- and high- Frequency|-|64.55|59.74|55.62|51.10|
> |Low- and high- Frequency|✓|65.64|60.67|56.39|52.44|
>
> **Q2**: Definition of causal refinement.
>
> In our work, causal refinement refers to the process of adjusting the frequency spectrum of feature representations to strengthen components associated with causal features (mid- frequency) and suppress those reliance to domain-invariant spurious factors (low- and high- frequency), thereby encouraging the model to rely more on generalizable features. While not an explicit causal model, the design is grounded in the theoretical research[1, 2] that mid-frequency components carry more causal information. Thus, our work translates this causal insight into a practical, trainable mechanism for DAOD.
>
> **Q3**: Whether the performance gains primarily come from the adaptive perturbation mechanism, or from the broader causal motivation itself?
>
> The performance gains in our framework stem from the integration of both the causal motivation and the adaptive perturbation mechanism, rather than from either component. Causal motivation provides the theoretical foundation and design principles, and the adaptive perturbation mechanism operationalizes this causal insight. Without the adaptive mechanism, even causally motivated perturbations remain static and cannot refine themselves over time, capping performance improvement, as evidenced by the suboptimal results of the fixed-band(**Line 370**).
>
> **Q4**: Comparison between framework design and existing patterns.
>
> We design a closed-loop system for dynamic adjustment of the data distribution, enabling targeted spurious factors perturbation and causal feature refinement. By shifting from a static, domain-invariant learning paradigm to an autonomous, adaptive causal feature learning paradigm, our work represents a paradigm shift for achieving robust object detection generalization. This moves the field beyond representational learning for statistical invariance toward causal feature learning.
>
> **Reference**
>
> [1]Multi-view Adversarial Discriminator: Mine the Non-causal features for Object Detection in Unseen Domains. CVPR, 2023.
>
> [2]Unbiased Faster R-CNN for Single-source Domain Generalized Object Detection. CVPR, 2024.
>
> [3]On the Spectral Bias of Neural Networks. ICML, 2019.
>
> [4]Linear CNNs Discover the Statistical Structure of the Dataset Using Only the Most Dominant Frequencies. ICML, 2023.
>
> [5]Towards Combating Frequency Simplicity-biased Learning for Domain Generalization. NeurIPS, 2024.
>
> [6]Frequency Shortcut Learning in Neural Networks. NeurIPSW, 2022.
>
> [7]What do Neural Networks Learn in Image Classification? A Frequency Chortcut Perspective. ICCV, 2023.

---

> > ### Author Rebuttal · Reviewer_7Twa · 2026-04-03
> >
> > The authors have addressed my concerns, so I will keep my original positive rating.

---

> > > ### Author Response · Authors · 2026-04-04
> > >
> > > Dear Reviewer 7Twa,
> > >
> > > Thank you for your acknowledgment of our response.  We are very happy to have addressed all your concerns and sincerely appreciate your recognition of our work.
> > >
> > > Thank you again for your time and constructive feedback.
> > >
> > > Sincerely,
> > >
> > > Authors of Submission 938

---

### Decision · Program_Chairs · 2026-04-30

**Decision:**

Accept (regular)

**Comment:**

The paper studies the well-known problem of domain-adaptive object detection and aims to address it via a causal refinement technique.
Among the main strengths mentioned by the reviewers are:

- builds around a relevant motivation
- well-presented idea and clearly explains each component
- results are SOTA and ablation studies are comprehensive
- design of DGCR module is interesting that dynamically adjusts the intensity of frequency perturbation

Some important weaknesses highlighted by the reviewers are:
- mismatch between causal motivation and actual implementation
- except dynamic closed loop adjustment, majority of other components are known components in literature
- sensitivity to initialization towards performance improvement claimed by core components
- limited to image modality
- difference with typical data augmentation mechanism

In the post-rebuttal phase, all reviewers agreed that their major concerns have been resolved, including potential misalignment between causal motivation and actual realization of it, robustness to initialization methods, and how the method could be extended to other modalities. As such, after the rebuttal acknowledgment, no reviewer mentioned of any partially resolved concerns and some of them even raised their scores. The final ratings for the paper are all weak accepts. Since there are no lingering issues or concerns in the submission left by any of the reviewers, the AC decides to recommend acceptance. Authors are encouraged to include important rebuttal and post-rebuttal discussion results in the final version.